# Uncertainty of the global oceanic $CO_2$ uptake induced by wind forcing: quantification and spatial analysis

Alizée Roobaert[1], Goulven G. Laruelle[1,2], Peter Landschützer[23], Pierre Regnier[1]

[1]Department Geoscience, Environment & Society (DGES), Université Libre de Bruxelles, Brussels, CP160/02, Belgium
[2]UMR 7619 Metis, Sorbonne Université, UPMC, Univ Paris 06, CNRS, EPHE, IPSL, Paris, France
[32]Max Planck Institute for Meteorology Bundesstr. 53, Hamburg, 20146, Germany

*Correspondence to*: Alizée Roobaert (Alizee.Roobaert@ulb.ac.be)

**Abstract.** The calculation of the air-water $CO_2$ exchange ($FCO_2$) in the ocean not only depends on the gradient in $CO_2$ partial pressure at the air-water interface but also on the parameterization of the gas exchange transfer velocity ($k$) and the
choice of wind product. Here, we present regional and global-scale quantifications of the uncertainty in $FCO_2$ induced by several widely used $k$-formulations and 4 wind speed data products (CCMP, ERA, NCEP1 and NCEP2). The analysis is performed at a 1° x 1° resolution using the sea surface $pCO_2$ climatology generated by Landschützer et al. (2015) for the 1991-2011 period while the regional assessment relies on the segmentation proposed by the Regional Carbon Cycle Assessment and Processes (RECCAP) project. First, we use $k$-formulations derived from the global [14]C inventory relying on
a quadratic relationship between $k$ and wind speed ($k = c \cdot U_{10}^2$, Sweeney et al., 2007; Takahashi et al., 2009; Wanninkhof, 2014) where $c$ is a calibration coefficient and $U_{10}$ is the wind speed measured 10 meters above the surface. Our results show that the range of global $FCO_2$, calculated with these $k$-relationships, diverge by 12 % when using CCMP, ERA or NCEP1. Due to differences in the regional wind patterns, regional discrepancies in $FCO_2$ are more pronounced than global. These global/regional differences significantly increase when using NCEP2 or other $k$-formulations which include earlier
relationships (i.e. Wanninkhof, 1992; Wanninkhof et al., 2009) as well as numerous local/regional parameterizations derived experimentally. To minimize uncertainties associated with the choice of wind product it is possible to recalculate the coefficient $c$ globally (hereafter called $c^*$) for a given wind product and its spatio-temporal resolution, in order to match the last evaluation of the global $k$ value. We thus performed these recalculations for each wind product at the resolution and time period of our study but the resulting global $FCO_2$ estimates still diverge by 10 %. These results also reveal that the
Equatorial Pacific, the North Atlantic and the Southern Ocean are the regions in which the choice of wind product will most strongly affect the estimation of the $FCO_2$, even when using $c^*$.

## 1 Introduction

Since the beginning of the industrial revolution, human activities such as fossil fuel burning, cement production and land use change have led to the increase of greenhouse gases concentrations in the atmosphere, altering the radiative balance of the
Earth system and changing the climate of our planet (IPCC, 2014). Current emissions of carbon dioxide ($CO_2$) exceed 10 Pg

C yr$^{-1}$ of which about half remains in the atmosphere (Le Quéré et al., 2016). The remainder is estimated to be taken up in roughly equal shares by the land and the ocean. In past decades, the magnitude of the ocean carbon sink was mainly estimated from global ocean biogeochemistry models and atmospheric inverse models but the recent increase in oceanic $CO_2$ measurements and the creation of the Surface Ocean $CO_2$ Atlas (SOCAT) database (Baker et al., 2014, 2016; Pfeil et al.,

2013; Sabine et al., 2013) has opened new research avenues, including the possibility to monitor the temporal evolution of the global oceanic carbon sink based on surface ocean $CO_2$ measurements (Landschützer et al., 2016; Rödenbeck et al., 2015). The exchange of $CO_2$ through the air-seawater interface can be estimated from the surface ocean $CO_2$ measurements using a relationship of the form:

$$FCO_2 = k \cdot K_0 \cdot \Delta pCO_2 \,, \tag{1}$$

where $k$ describes the wind driven kinetic gas transfer of $CO_2$ between the ocean and the atmosphere, $K_0$ is the sea surface temperature and salinity dependent solubility of $CO_2$ and $\Delta pCO_2$ describes the measured partial pressure difference between the ocean and the atmosphere. Observationally-based flux estimates suggest a substantially weaker ocean $CO_2$ uptake compared to models and inverse analyses (Wanninkhof et al., 2013a). While the increasing number of measurements and recent improvement in data-interpolation techniques (e.g. Landschützer et al., 2014; Laruelle et al., 2017; Rödenbeck et al.,

2013; Sasse et al., 2013) help to better constrain the $\Delta pCO_2$ factor, previous studies (Landschützer et al., 2014; Takahashi et al., 2009) further suggest that a large source of uncertainty in the ocean $CO_2$ uptake stems from the quantification of the gas transfer velocity $k$. In the past, $k$ has been estimated in the laboratory from wind tunnel studies (e.g. Liss and Merlivat, 1986) and in the field using several methods such as tracer measurements (e.g. Ho et al., 2006) and eddy covariance methods (e.g. Prytherch et al., 2010). While all existing parametrizations of $k$ find a strong relationship with the wind speed 10 meters

above sea surface ($U_{10}$), a wide variety of formulations have been proposed. In the literature, relationships between $k$ and $U_{10}$ include linear (e.g. Liss and Merlivat, 1986), quadratic (e.g. Wanninkhof, 1992), cubic (e.g. Wanninkhof and McGillis, 1999), a combination of linear and quadratic (e.g. Weiss et al., 2007) as well as a combination of linear, quadratic and cubic formulations (Wanninkhof et al., 2009). Because of the quadratic or cubic components involved in most of those parametrizations, the differences in $k$ estimates are generally small in the low to mid wind speed range but substantially

increase when high wind speed regimes are considered (e.g. Woolf, 2005).

Few studies calculating the global oceanic carbon sink from surface ocean $CO_2$ measurements have tried to quantify the uncertainty associated to with the variety of existing gas transfer formulations (Landschützer et al., 2014; Sweeney et al., 2007; Takahashi et al., 2009; Wanninkhof and Trinanes, 2017) and those who do have only used a subset of existing $k$-

formulations forced by a single wind field (Landschützer et al., 2014; Wanninkhof and Trinanes, 2017). Despite these limitations, past estimates suggest that a substantial amount of uncertainty, in the range 30-37 % of the mean global ocean carbon uptake, could arise from $k$. Yet no study has to-date fully assessed the effect of using different wind products and $k$-formulations on the global air-sea exchange of $CO_2$ and its spatial variability. Here we provide a detailed quantification of

air-sea $CO_2$ fluxes considering the most commonly established *k*-parametrizations and four widely used wind products. We then perform an extensive assessment of global and regional flux uncertainty estimates to help better constrain the ocean carbon uptake based on observations. In particular, we provide the first wind-induced uncertainty estimate of the ocean ~~$FCO_2$~~-latitudinal distribution of $FCO_2$ at the global scale. This analysis is particularly relevant for global carbon budget analysis (Le Quéré et al., 2016; Sarmiento et al., 2010) since to-date the quantification of the global land sink is still largely dependent on the quantification of the ocean carbon uptake.

## 2 Methodology

### 2.1 Formulation of $CO_2$ gas transfer at the air-sea interface

The theoretical background of the gas transfer is well established and extensively described in Deacon (1977), Liss and Merlivat (1986) and in Sarmiento and Gruber (2006). ~~In a nutshell, g~~Gas transfer that occurs at the air-water interface, $FCO_2$ [mol C m$^{-2}$ yr$^{-1}$], in a micrometric water and air boundary layer, can be estimated by Fick's first law of molecular diffusion:

$$FCO_2 = D \cdot \frac{\partial c}{\partial z} , \tag{2}$$

where $D$ is the molecular diffusion coefficient of $CO_2$ [m$^2$ yr$^{-1}$] and $z$ [m] the liquid and gas film thickness. Since the concentration gradient is difficult to measure ~~because~~ as $z$ is very small (Blade, 2010), gas exchange transfer is often expressed as in Liss and Merlivat (1986):

$$FCO_2 = k_{tot} \cdot \Delta CO_2 , \tag{3}$$

where $\Delta CO_2$ [mol m$^{-3}$] represents the difference in $CO_2$ concentration between air and water and $k_{tot}$ is the gas transfer velocity of $CO_2$ [m yr$^{-1}$]. Following Henry's law and considering that transfer is only limited in the liquid layer because it is two orders magnitude slower than the transfer in the air layer (Sarmiento and Gruber, 2006), $FCO_2$ can be expressed in terms of partial pressure rather than concentration:

$$FCO_2 = k \cdot K_0 \cdot \Delta pCO_2 , \tag{4}$$

where $K_0$ is the aqueous-phase solubility of $CO_2$ in water [mol m$^{-3}$ atm$^{-1}$], which depends on the sea surface temperature (*SST*) and salinity (*SSS*) and is calculated following Weiss (1974), and $\Delta pCO_2$ represents the partial pressure difference between $pCO_2$ in the ocean ($pCO_{2,water}$, referred to as $pCO_2$ in what follows) and in the atmosphere ($pCO_{2,air}$) [atm]. By convention, and following the sign of the $pCO_2$ gradient, negative values of $FCO_2$ correspond to a transfer of $CO_2$ from the atmosphere to the ocean (i.e. a sink for the atmosphere) and positive values of $FCO_2$ correspond to a transfer of $CO_2$ from the ocean to the atmosphere (i.e. a source for the atmosphere). The gas transfer velocity of $CO_2$ in the liquid layer ($k$) depends on the molecular diffusivity (which is a function of *SST* and *SSS*) as well as on the hydrodynamics of the aqueous phase and the characteristics of the diffusion layer (Wanninkhof et al., 2009). In order to isolate the influence of the

hydrodynamics within the water layer, that is to say, the turbulence at the interface, $k$ is normalized to a Schmidt number ($Sc$) of 660, which represent the gas exchange transfer velocity of $CO_2$ at 20°C in seawater ($SSS = 35$):

$$k_{660} = k_{SST,SSS} \cdot \left(\frac{Sc_{SST,SSS}}{660}\right)^{1/2} \qquad (5)$$

The value of the exponent 1/2 is experimentally derived (Jähne et al., 1987) and corresponds to conditions of a wavy rough
surface representative of the oceanic sea surface (Wesslander et al., 2011). Combining Eq. (4) and (5) leads to the following formula for the $CO_2$ exchange at the air-sea interface:

$$FCO_{2SST,SSS} = k_{660} \cdot K_0 \cdot (1 - Ice) \cdot \Delta pCO_2 \cdot \left(\frac{Sc_{SST,SSS}}{660}\right)^{-1/2} \qquad (6)$$

Where $FCO_2$ is expressed in mol C m$^{-2}$ yr$^{-1}$, $k_{660}$ in m yr$^{-1}$, $K_0$ in mol m$^{-3}$ atm$^{-1}$ and $\Delta pCO_2$ in atm. $Sc$ (dimensionless) is calculated according to the equation reported by Wanninkhof (2014). $Ice$ represents the fraction of the ocean covered by sea
ice (comprised between 0 for ice free and 1 for entirely covered), which is assumed to inhibit the air-sea $CO_2$ transfer (Evans et al., 2015; Landschützer et al., 2013; Laruelle et al., 2014).

## 2.2 Data products

We use a 21 year observationally-based global monthly gridded sea surface $pCO_2$ product covering the 1991 through 2011 period (Landschützer et al., 2015). This period was chosen to cover the overlapping temporal extent of the four wind
products selected for this study. For our analysis we create a climatological monthly mean $FCO_2$ estimate from our gridded $pCO_2$ fields over this period and atmospheric partial pressures of $CO_2$ ($pCO_{2,air}$) calculated from the NOAA Marine Boundary Layer reference product at 100 % humidity (Dickson et al., 2007). The $pCO_2$ fields are based on measurements of the Surface Ocean $CO_2$ Atlas version 2 (SOCATv2) dataset (Baker et al., 2014) using a two--steps artificial neuronal network (Landschützer et al., 2015) to generate continuous monthly 1° x 1° resolution $\Delta pCO_2$ maps for the global ocean excluding
the Arctic Ocean, coastal regions and marginal seas. A more detailed description of the method and its extensive evaluation can be found in Landschützer et al. (2013, 2014, 2016). Four global wind speed datasets are used to evaluate the sensitivity of $FCO_2$ to the choice of one wind product over the other. The four data products selected are the most widely used in the literature: Cross-Calibrated Multi-Platform Ocean Wind Vector 3.0 (CCMP, Atlas et al., 2011), the global atmospheric reanalysis ERA-interim (ERA, Dee et al., 2011), the NCEP/NCAR reanalysis 1 (NCEP1, Kalnay et al., 1996) and the
NCEP/DOE AMIP-II Reanalysis (NCEP2, Kanamitsu et al., 2002). The latter is an update of NCEP1, using an improved forecast model and data assimilation system (Kanamitsu et al., 2002). To achieve the same 1° x 1° spatial resolution for the wind field as that of $\Delta pCO_2$, a cells aggregation is performed for CCMP and ERA that have finer spatial resolutions (0.25° x 0.25°). This aggregation generates a 1° x 1° grid by performing surface weighted averages of all the wind speed values comprised in each 1° x 1° cell. The original spatial resolution of both NCEP1 and NCEP2 is a global T62 Gaussian grid (i.e.
192 longitudes equally spread and 94 latitudes unequally spread) and is translated into a continuous 1° x 1° data field using a

two-dimensional spline interpolation. The original spatial resolutions of the four wind speed products are summarized in Table 1. Their temporal resolution is the same (6 hours) and much finer than the one of $\Delta pCO_2$. Therefore, centered monthly mean for the wind speed ($<U_{10}>$) and its second moment ($<U_{10}^2>$) are calculated to match the temporal resolution of the $\Delta pCO_2$ data. The use of $<U_{10}^2>$ allows accounting for the variance of wind speed in the $k$ estimates. In what follows,

$FCO_2$ calculated with these different wind datasets are referred to as $FCO_{2\text{-}CCMP}$, $FCO_{2\text{-}ERA}$, $FCO_{2\text{-}NCEP1}$ and $FCO_{2\text{-}NCEP2}$, respectively. All the calculations are performed using the $CO_2$ solubility ($K_0$) product calculated by Landschützer et al. (2015) following Weiss (1974), the sea ice fraction from Rayner et al. (2003) and the sea surface temperature ($SST$) from the NOAA OI SST V2 (daily 0.25° x 0.25° resolution, Reynolds et al., 2007). We transformed the original $SST$ data to monthly mean values at 1° spatial resolution following the same procedure as that used for the CCMP wind data. The Schmidt

number was calculated using the transformed $SST$ field and the equation proposed by Wanninkhof (2014). Last, the boundaries of the domain of calculations correspond to the land-sea mask from Landschützer et al. (2015), which covers 317.7 $10^6$ km$^2$ of the open ocean area, omitting the Arctic Ocean, coastal regions and marginal seas.

### 2.3 $k\text{-}U_{10}$ Parameterization

In the open ocean, wind stress is the dominant hydrodynamic factor controlling the level of turbulence at the air-sea interface

and thus is the key control factor of $k$ (Sarmiento and Gruber, 2006). As reported in Table 2, all studies agree with the concept that $k$ can be parametrized by a function of wind speed to the power of $n$, with $n \geq 1$. This dependency was demonstrated empirically in a number of local and regional experimental studies, using diverse methods such as covariance flux or deliberate tracers (i.e. helium ($^3$He) and sulfur hexafluoride ($SF_6$)) techniques. These studies have led to $k$ dependencies on wind speed of quadratic ($k_{660} = c \cdot U_{10}^2$, Ho et al., 2006, 2011; Jacobs et al., 1999; Kuss et al., 2004), cubic

($k_{660} = a+d \cdot U_{10}^3$ with a $\geq$ 0, Edson et al., 2011; Kuss et al., 2004; McGillis et al., 2001, 2004; Prytherch et al., 2010; Wanninkhof and McGillis, 1999) and linear-quadratic ($k_{660} = b \cdot U_{10}+c \cdot U_{10}^2$, Nightingale et al., 2000; Weiss et al., 2007) forms. Other studies have followed a distinct approach and constrained $k\text{-}U_{10}$ relationships for the global ocean on the basis of the global ocean bomb $^{14}$C inventory (Broecker et al., 1985; Naegler et al., 2006; Sweeney et al., 2007) and global wind fields. The resulting relationships are all of quadratic form ($k_{660}= c \cdot U_{10}^2$), with different global values of $c$ depending on the

spatio-temporal resolution of the wind speed product used. Therefore, in principle, values reported for $c$ in Table 2 are intimately associated to the specific wind product that was applied during the fitting procedure (Naegler et al., 2006). For further details regarding the different procedures, refer to Table 2. Note that the $k\text{-}U_{10}$ relationships only hold for a range of wind values as they were constrained from observations performed within a range of wind speed conditions (Fig. 1), in particular for empirical approaches.

**2.4 Sensitivity and uncertainty analysis**

The uncertainties in the air-sea exchange of $CO_2$ arising from wind products and $k\text{-}U_{10}$ formulations are assessed at the 1° x 1° resolution over the 1991-2011 period. As a first step, the effect of the chosen wind product is investigated alone and

global and regional $FCO_2$ are calculated using the latest $k$-parameterization proposed by Wanninkhof (2014). In a second step, we calculate regionally and globally integrated $FCO_2$ using a given wind product combined with different global $k$-$U_{10}$ formulations derived from [14]C bomb inventories (equations in bold in Table 2). Here, empirical relationships derived from local and regional studies are not used because they were not calibrated for the global wind products applied in our study and are generally designed for specific local conditions (i.e. Jacobs et al., 1999; Kuss et al., 2004; Weiss et al., 2007). Although Wanninkhof (2014) recently proposed a new value for the $c$ coefficient (0.251), we also used the value proposed by Wanninkhof (1992, $c = 0.31$) in our analysis as it is still widely used in global and regional $FCO_2$ studies (e.g. Aumont and Bopp, 2006; Bourgeois et al., 2016; Matear and Lenton, 2008; Le Quéré et al., 2007; Schwinger et al., 2016; Thomas et al., 2008). In addition to the four quadratic equations obtained from [14]C bomb inventories, we also included the hybrid equation of Wanninkhof et al. (2009) (also identified in bold in Table 2) since it is applicable to the entire range of wind speeds encountered in the ocean and was developed from a literature review of global scope. Although they were not included in our quantitative uncertainty estimate, a global $FCO_2$ calculation was also performed using 6 empirical $k$-relationships with different dependencies on $U_{10}$. For each functional relationship, we choose one or two formulations that are applicable over the entire range of wind speeds reported for the oceanic surface. The selected cubic form ($k_{660} = d \cdot u_{10}^3$) is the one by Kuss et al. (2004) while the $k_{660} = a + d \cdot u_{10}^3$ form is constrained by the Prytherch et al. (2010) and Edson et al. (2011) parameterizations. We used the Weiss et al. (2007) relationship for the $k_{660} = b \cdot U_{10} + c \cdot U_{10}^2$ formulation, and those reported by Kuss et al. (2004) and Ho et al. (2011), with a recalculation of the coefficient $c$ for $k_{660}$ for the quadratic form ($k_{660} = c \cdot U_{10}^2$). The Kuss et al. (2004) and Ho et al. (2011) parameterizations were selected to provide upper and lower bound estimates for the quadratic formulations. Finally, a latitudinal and regional assessment of $FCO_2$ is performed using the regions defined within the ocean Regional Carbon Cycle Assessment and Processes (RECCAP) program (Canadell et al., 2011), which follow regions designed to analyze atmospheric inversions data (Gurney et al., 2008). In this context, the ocean is sub-divided into 11 regions: North Pacific (1), Equatorial Pacific (2, 3), South Pacific (4), North Atlantic (5, 6), Equatorial/South Atlantic (7, 8), Southern Ocean (9) and North/South Indian Ocean (10, 11). These regions were created with the aim of obtaining, on the basis of interdisciplinary and independent studies, an improved knowledge of regional carbon sources and sinks estimates and the underlying processes involved.

## 3 Results

### 3.1 $FCO_2$ uncertainty arising from the choice of wind product

The zonal mean air-sea $FCO_2$ (mol C m$^{-2}$ yr$^{-1}$) calculated using the four wind speed datasets is illustrated in Fig. 2a. The four mean latitudinal profiles reveal strong qualitative similarities that reflect both the latitudinal $U_{10}$ (Fig. 2b) and $\Delta pCO_2$ distributions (Fig. 2c). In the Northern Hemisphere, high latitudes (> 40° N) acts as strong $CO_2$ sinks while the tropics are close to neutral and a narrow latitudinal band around the Equator is a moderate $CO_2$ source. In the Southern Hemisphere, a strong $CO_2$ sink can be observed at around 40° S while the Southern Ocean further south is quasi neutral. In quantitative

terms, substantial differences between profiles can however be observed, especially in the equatorial and mid latitudes as a result of differences in the applied wind speed products.

The vast majority of the monthly averaged wind speeds fall in the 3 m s$^{-1}$ to 8 m s$^{-1}$ range at low and intermediate latitudes (Fig. 2b) but stronger winds are often observed at high latitudes. Wind speeds above 8 m s$^{-1}$ are mainly located within the ~ 40-60° N and S latitudinal band where the $pCO_2$ gradient is negative (Fig. 2c), resulting in the strong $CO_2$ sink regions of the global ocean. Climatological mean wind speeds exceeding 10 m s$^{-1}$ are rare and only occur in the Austral Ocean, where the $pCO_2$ gradient is close to equilibrium or slightly positive; hence, despite the fast gas transfer, the exchange of $CO_2$ is small. Table 1 compares the global average climatological mean wind speeds from the four products used in this study. On average, the highest global mean wind speed is generated using NCEP2 (8.2 m s$^{-1}$) and the lowest using NCEP1 (7.2 m s$^{-1}$). This is also reflected in the latitudinal $U_{10}$ profiles of Fig. 2b. The wind distribution for CCMP and ERA are much similar to NCEP1 than to NCEP2 and a marked difference is thus observed between NCEP2 and the other wind products. The effect of varying wind speeds is further illustrated in Fig. 3, which compares the 21 year mean oceanic air-sea $FCO_2$ maps using the two wind products yielding very contrasting $FCO_2$ maps (NCEP1 and NCEP2, respectively). Not surprisingly, the trends displayed in Fig. 3 are both consistent with Fig. 2a and previous research (e.g. Landschützer et al., 2013, 2015; Takahashi et al., 2009) and reveal that the calculated $FCO_2$ is generally positive around the equatorial upwelling regions and in the Austral Ocean (50°-70° S). Along the tropics (23° N and S) and in the high latitudes, the ocean behaves as a sink for $CO_2$, with the notable exception of the coastal regions. Figure 3 shows that the $FCO_2$ calculated for each 1° x 1° cell using NCEP2 is larger than that obtained using NCEP1 over 87 % of the oceanic surface area. In addition, computing the flux with NCEP1 leads to the lowest $FCO_2$ for 52 % of the oceanic surface area compared to all other wind datasets. The discrepancies between $FCO_2$ generated using NCEP2 and those generated using the other wind products are particularly pronounced near the equator, in the Arctic region and around 40° S (Austral Ocean) and 40° N (Fig. 2a and Fig. 3). For example, at these mid-latitudes in the north and south hemisphere, differences between $FCO_{2\text{-}NCEP1}$ and $FCO_{2\text{-}NCEP2}$ can reach 0.8 and 0.6 mol C m$^{-2}$ yr$^{-1}$, respectively. Such pronounced differences result from the combination of relatively high wind speeds and significant $pCO_2$ gradients (> 25 μatm) as well as significant discrepancies between NCEP1 and NCEP2 at these latitudes (Fig. 2b). Other regions characterized by large differences in $FCO_2$ depending on the applied wind product include western boundary currents such as the Brazilian/Malvinas Current and the Florida Current, which generally are regions of intense $CO_2$ outgassing (Cai, 2011; Laruelle et al., 2010, 2014). It should be noted, however, that the spatial extent of our $pCO_2$ data product does not include the near coastal zone and thus only partly cover these areas. Comparing the air-sea $CO_2$ exchange using all climatological mean wind products, we find that CCMP (global wind average of 7.5 m s$^{-1}$ from 1991 through 2011, which is close to that calculated by Wanninkhof (2014) for the period 1990-2009 of 7.3 m s$^{-1}$) leads to a slightly more intense $CO_2$ exchange between 40° S-40° N and in the Arctic region (> 60° N) than $FCO_{2\text{-}ERA}$ and $FCO_{2\text{-}NCEP1}$ (Fig.2a) The differences between the median $FCO_2$ fields generated using ERA and NCEP1 are very small (< 0.1 mol C m$^{-2}$ yr$^{-1}$) and either wind product can yield the most intense $FCO_2$ from a region to the other. In what follows, we compare the

climatological mean $U_{10}$ and air-sea $FCO_2$ (Tg C yr$^{-1}$) for the 11 ocean RECCAP regions (Canadell et al., 2011; see Table S1 in the supplement). The results expressed in percentage correspond to the relative difference between the highest and lowest $FCO_2$ values for a given region and are synthetized in Table 3.

Overall, the relative differences between average wind speed fall in the 10-16 % range across the 11 RECCAP regions (Table S1), which translate into relative variations in $FCO_2$ ranging from 21-30 %, except in the Equatorial Pacific (region 2) where variability reaches 42 %. It should be noted that in this region the $pCO_2$ is close to atmospheric levels and, thus, this high relative variability does not translate into large absolute differences in $FCO_2$. Hence, the relative percentage difference of regions 2 and 3 will be expressed in one region only. Most of the variability results from the use of the NCEP2 dataset,
especially in regions of high wind speeds. Excluding NCEP2, the relative variability in $FCO_2$ drops below 10 % for all regions, except in the Equatorial Pacific (regions 2 and 3).

Globally, the 21 year average oceanic $CO_2$ flux calculated with the different wind products varies between -1.30 Pg C yr$^{-1}$ and -1.38 Pg C yr$^{-1}$ using ERA, NCEP1 and CCMP in conjunction with the formulation of $k$ proposed by Wanninkhof (2014,
Table 4). These estimates are thus consistent with each other, but fall in the low end of the range of global estimates published for the global oceanic $CO_2$ uptake (Gruber et al., 2009; Takahashi et al., 2009; Wanninkhof et al., 2013b). This result can partly be explained by the absence of the Arctic Ocean and coastal regions in our $pCO_2$ climatology (Landschützer et al., 2014). Using the NCEP2 dataset and the Wanninkhof (2014) formulation, the $FCO_2$ increases significantly to -1.84 Pg C yr$^{-1}$. In relative terms, this represents a difference of about 29 % in the global ocean $CO_2$ uptake estimate across all wind
products (Table 3). Thus, even with the use of the same $k$-$U_{10}$ equation, the choice of wind products can lead to significantly different global and regional $FCO_2$ estimates.

### 3.2 $FCO_2$ uncertainty arising from global $k$-$U_{10}$ parameterizations and wind products

In this section, we constrain the uncertainty in air-sea $CO_2$ fluxes associated to the use of all published $k$-$U_{10}$ parameterizations derived from [14]C bomb inventories and the hybrid formulation of Wanninkhof et al. (2009) (Tables S1 and
4). We also report estimates obtained with the Wanninkhof (1992) formulation in these tables, but exclude it from our analysis as it is now accepted that this parameterization is outdated (Wanninkhof, 2014). Globally, we find that the 21 year mean ocean uptake of $CO_2$ averaged across all quadratic formulations and wind speed datasets is -1.52 Pg C yr$^{-1}$ (Table 4), in agreement but again on the lower end of previous estimates (Gruber et al., 2009; Takahashi et al., 2009; Wanninkhof et al., 2013b). However, the range is significant and varies from -1.30 to -1.98 Pg C yr$^{-1}$. This range is even larger (-1.19 to -1.98
Pg C yr$^{-1}$) when the hybrid formulation is also included in the analysis. The 40 % global $FCO_2$ relative uncertainty mainly stems from the use of the NCEP2 wind product (Table 3). Yet, even without NCEP2, the resulting uncertainties of 12 and 20 % for the quadratic only and quadratic-hybrid parameterizations, respectively, are still significant. As reported in Table 4,

the global $FCO_2$ estimates for different $k$-formulations combined with a single wind product show roughly similar relative differences around 13 % (Table 3).

The spatial distribution of $FCO_2$ corresponding to the minimum (Wanninkhof et al., 2009 with NCEP1) and maximum (Sweeney et al., 2007 with NCEP2) global ocean $CO_2$ uptake are illustrated in Fig. 4. Results reveal that the difference in flux intensity between the two estimates is significant, particularly in the Equatorial Pacific and the mid/high latitudes, where the strongest $pCO_2$ gradients are identified. The spatial patterns in $FCO_2$ are further investigated by aggregating the results at the regional scale of the RECCAP regions. Using the same wind speed product, the relative differences between $FCO_2$ estimated with the various quadratic $k$-relationships never exceed 7 %. In general, we find that the smallest $FCO_2$ uptake is obtained with the hybrid formulation of $k$. Thus, including the quadratic-hybrid formulations, the relative difference in $FCO_2$ estimates for a given wind speed product increase to 7-16 % (Table 3). Overall, the range of estimates is now much larger and the relative differences reach 27-40 % across RECCAP regions. These uncertainties are slightly reduced (maximum relative uncertainty of 35 %) if the hybrid formulation is excluded. When only using quadratic equations, we find the largest flux uncertainties in the Equatorial Pacific (regions 2 and 3), North Atlantic (region 5) and Southern Ocean (region 9) where the variations in wind speed estimates are also the largest (Table S1). The Equatorial Pacific is the largest source region of the open ocean and our estimated range is comprised between 0.34 and 0.53 Pg C yr$^{-1}$ (region 2 and 3). In contrast, the North Atlantic (regions 5 and 6) and Southern Ocean are important sink regions for which estimates fall in the 0.33-0.53 and 0.22-0.37 Pg C yr$^{-1}$ ranges, respectively.

### 3.3 $FCO_2$ estimates using empirical $k$-parameterizations

For comparison, global oceanic $FCO_2$ were also calculated using 6 empirical $k$-relationships (Table 4). Overall, and regardless of the wind product used, the global $FCO_2$ sinks predicted using these formulations are significantly larger than those based on the global ocean bomb $^{14}$C inventory, with the notable exception of the formulation proposed by Ho et al. (2011) that was derived from an extensive collection of data sampled in different locations. All other empirical relationships tested for the analysis are derived from local or regional studies and yield global $FCO_2$ estimates ranging from -2.07 Pg C yr$^{-1}$ using the linear and quadratic formulation of Weiss et al. (2007) in conjunction with ERA to -4.2 Pg C yr$^{-1}$ using the cubic relationship of Kuss et al. (2004) in conjunction with NCEP2. This corresponds to a 2 fold increase in the global $FCO_2$ estimate despite the fact that these empirical formulations were derived from measurements performed in the same region (Baltic Sea).

### 4 Discussion

In the ocean, a vast literature has been published on the parameterization of $k$ over the past 25 years (Table 2). At the global scale, the parametrization of $k$ in $FCO_2$ follows a quadratic form ($k_{660} = c \cdot U_{10}^2$) and is performed by constraining the

coefficient $c$ using the spatio-temporally integrated $^{14}C$ bomb inventory as described in Sweeney et al. (2007). In essence, this method computes a single global value of $c$ to match the observed evolution over time of the global oceanic stock of the radiotracer $^{14}C$, which results from its invasion through the air-water interface. All but the hybrid and the empirical $k$-relationships used here have been constrained using this concept (Sweeney et al., 2007; Takahashi et al., 2009; Wanninkhof,

1992, 2014). This approach was first proposed by Wanninkhof (1992) based on the $^{14}C$ global inventory estimated by Broecker et al. (1985). Since then, the $^{14}C$ inventory in the ocean has been reassessed (Naegler et al., 2006; Sweeney et al., 2007), new spatially resolved global wind products have been released and, therefore, the original coefficient $c$ calculated by Wanninkhof (1992) has been repeatedly updated (Sweeney et al., 2007; Takahashi et al., 2009; Wanninkhof, 2014). In particular, Naegler et al. (2006) have shown that the value of $c$ is a function of the applied wind field but also of its spatio-

temporal resolution. Hence, in principle, the selection of a given $c$ value not only implies the use of the same wind speed product as the one originally applied, but also retention of the same spatio-temporal resolution and temporal coverage. If another wind product and/or different spatial and temporal resolutions are used to calculate $FCO_2$, then the value of $c$ has to be adapted accordingly. Naegler et al. (2006) proposed different correction coefficients for $c$ in order to account for the bias introduced by the choice of one wind product over another. These correction coefficients have been calculated for several

sets of spatial and temporal resolutions (1° x 1°, daily or 5° x 4°, monthly for example). Although useful, the coefficients calculated by Naegler et al. (2006) would now need to be updated to comply with new estimates of the global $^{14}C$ inventory (Sweeney et al., 2007) and new combinations of spatial and temporal resolution. Another approach to calibrate the value of the coefficient $c$ consists of recalculating its value (called hereafter $c*$) to match the latest globally averaged value of $k$ taken from the literature (Naegler, 2009; Sweeney et al., 2007; Wanninkhof et al., 2013b) over a given period using the wind

product and its associated resolution (e.g. Landschützer et al., 2014). However, this method is only suitable for global calculations, and can thus not be applied to regional or smaller scale studies. The use of the two above methods is far from being standard procedure yet and only a few studies have adapted their coefficient $c$ prior to calculating global or regional $FCO_2$. In numerous modeling studies, $FCO_2$ is still calculated using $k$-parametrizations from the literature combined with a different wind product from the one used to calibrate the coefficient $c$ (e.g. Aumont and Bopp, 2006; Bourgeois et al., 2016;

Matear and Lenton, 2008; Le Quéré et al., 2007; Schwinger et al., 2016; Thomas et al., 2008). These inconsistencies call into question the assessment of the wind-induced uncertainties associated with the future ocean $CO_2$ sink and a systematic approach, similar to the one used for the observation-based estimation of the present-day $FCO_2$, should help better constrain model-derived uncertainties.

The calculations performed in this study allow quantification of the differences between $FCO_2$ estimates obtained using different wind products combined with quadratic $k$-relationships from the literature (Sweeney et al., 2007; Takahashi et al., 2009; Wanninkhof, 2014) without recalibration of their coefficient $c$. Our results indicate that, globally, the application of the ERA, CCMP or NCEP1 wind speed products only leads to small differences (~ 0.08 Pg C yr$^{-1}$, 6 % difference) when the same quadratic formulation is used (Tables 3 and 4). In addition, different $k$-parametrizations for a given wind speed product

induce differences in $FCO_2$ about twice larger than those associated with the choice of wind product itself. Overall, the combined effect of the three wind products and three quadratic $k$-formulations leads to a 0.18 Pg C yr$^{-1}$ (12 %) difference in global $FCO_2$. The hybrid formulation generally yields lower $FCO_2$ and, therefore, this range is extended to 0.29 Pg C yr$^{-1}$ (20 %) when this formulation is also included. The significant discrepancies between the NCEP2 and other wind products

translate into larger differences in $FCO_2$, especially in regions characterized by high wind speeds. This result is consistent with the findings of Winterfeldt and Weisse (2008) which report wind speeds up to 1.5 m s$^{-1}$ faster with NCEP2 than NCEP1 over some oceanic regions. The authors attribute these differences to changes in the parameterization of the convection scheme, leading to more intense storms. Wallcraft et al. (2009) also conclude that NCEP2 is inconsistent in magnitude and wind pattern over the ocean compared to the others products. Thus, although being an updated version of NCEP1, NCEP2 is

a multi-layer atmospheric wind product that provides better wind speed estimates overall but is not necessarily more accurate at sea-surface level (Hong and Pan, 1996).

Because wind patterns differ from one product to another, distinct combinations of $k$-formulations and wind speed products yielding the same global $FCO_2$ value (for instance Wanninkhof (2014) with CCMP and Takahashi et al. (2009) with ERA)

may lead to different $FCO_2$ estimates at the regional scale (Tables S1 and 4). These differences are most pronounced in the Equatorial Pacific (regions 2, 3). Ishii et al. (2014) used an ocean biogeochemistry model relying on the same quadratic $k$-parametrization for the $CO_2$ exchange to quantify $FCO_2$ for the Pacific Ocean using NCEP1 and CCMP. They obtained 0.22, 0.09 and 0.13 Pg C yr$^{-1}$ differences between both products (1990-2008 period) for the Equatorial, North and South Pacific, respectively. Consistent with our results, CCMP consistently led to a more intense $FCO_2$. Similar to the global scale results,

significantly larger differences in regional $FCO_2$ estimates are obtained when combining the different quadratic $k$-parametrizations with ERA, CCMP and NCEP1 and these discrepancies are amplified when including the hybrid formulation of Wanninkhof et al. (2009) and NCEP2 in the analysis (Table 3). Such differences result from the combination of different regional wind patterns, which might not be equally resolved by the different wind products (Table S1).

Besides the main global formulations discussed above, one also finds in the published~~the~~ literature ~~also reports~~ numerous empirical relationships mostly derived from local experiments. These formulations assume different functional relationships to the wind forcing (Table 2) and, when applied globally, yield widely contrasting results with generally much lower $FCO_2$ than the globally derived formulations (Table 4). This further supports the idea that empirical formulations are calibrated for specific local settings and are not suitable for global scale applications. For instance, the Kuss et al. (2004) and the Weiss et

al. (2007) relationships were derived in areas of the Baltic Sea characterized by very high wind speeds, up to 20 m s$^{-1}$. In addition, locally, wind may influence the intensity of the $CO_2$ exchange at the air-water interface by other processes not connected to the turbulence at the interface and the piston velocity. Rodgers et al. (2014), for example, identified the effect of wind speed as a control of $FCO_2$ in the Southern Ocean through its control on the depth of the mixed layer depth through wind stirring. This kind of indirect controls of wind on the $CO_2$ exchange at the air-water interface adds an additional

important source of uncertainty on local parameterization of $k$. Moreover, distinct methods have been applied to quantify $FCO_2$ experimentally. For instance, Weiss et al. (2007), Prytherch et al. (2010) and Edson et al. (2011) used the eddy covariance method while Ho et al. (2011) used the tracer method to determine $k$ empirically. These two methods have their relative advantages ~~and disadvantages~~ (see e.g. Garbe et al., 2014 for a review) but it has been ~~showed~~ shown that $k$ measured by the eddy covariance method leads to higher values than other methods (e.g. Jacobs et al., 2002). The only empirical formulation of $k$ that could eventually be applied globally is that of Ho et al. (2011) due to the variety of data used for calibration. The large discrepancies between global $FCO_2$ estimates calculated using global and empirical formulations highlights the importance of local phenomena such as bubble formation, extreme winds, fetch or the presence of surfactants at the sea surface, which affect the $CO_2$ exchange at the air water interface.

The differences between global $FCO_2$ calculated using a quadratic $k$-formulation where $c$ is rescaled ($c*$) for each of the four wind products allows us to constrain more accurately the effect of the chosen wind product. For each wind product, calculations are performed at a 1° x 1° resolution using 6 hourly $<U_{10}^2>$ fields to match a global average $k$ value of 16 cm h$^{-1}$ (Wanninkhof et al. 2013b; global mean $k$ of 15.95 cm h$^{-1}$ using CCMP) for the 1991-2011 period investigated here. Values of $c*$ equal to 0.271, 0.279, 0.211 of are obtained for ERA, NCEP1 and NCEP2, respectively. A $c*$ value of 0.256 is obtained for CCMP, which is close to the value of 0.254 calculated by Landschützer et al. (2014) for the 1998-2011 period and 0.251 calculated by Wanninkhof (2014) for the 1990-2009 period. The use of $c*$ and their corresponding wind products (Tables 3 and 4) leads to a 0.16 Pg C yr$^{-1}$ difference in global $FCO_2$. Our results therefore indicate that rescaling the $c$ coefficients considerably reduces the differences in global $FCO_2$ estimates, but the choice of a given pair of $c*$-wind product still yields uncertainties that can reach 11 % (compare $FCO_{2-CCMP}$ and $FCO_{2-NCEP2}$). This difference is comparable to that reported by Wanninkhof (2014) (0.2 Pg C yr$^{-1}$ or 15 %) using the $\Delta pCO_2$ climatology of Takahashi et al. (2009) over the 1990-2009 period. The $FCO_2$ integrated over the different RECCAP regions calculated with each wind product as well as with $c*$ are reported in Fig. 5 (see also Table S1). While differences remain limited under most latitudes for all oceanic regions (< 10 %), relative differences in regional $FCO_2$ estimates exceed 10 % in the Equatorial Pacific (regions 2 and 3, 17 % or 77 Tg C yr$^{-1}$), North Atlantic (region 5, 10 % or 20 Tg C yr$^{-1}$) and in the Southern Ocean (region 9, 14 % or 41 Tg C yr$^{-1}$). Therefore, the recalibration of $c$ to a given wind product at a specific spatio-temporal resolution considerably reduces the differences in $FCO_2$ estimates at the scale of the RECCAP regions (Tables S1 and 3).

In what follows, we compare our regional $FCO_2$ (Tg C yr$^{-1}$) calculated with $c*$ for the different oceanic basins to the results compiled in the RECCAP project (Ishii et al., 2014; Lenton et al., 2013; Sarma et al., 2013; Schuster et al., 2013) and in Zscheischler et al. (2017). The $FCO_2$ mean values and associated uncertainties reported in the RECCAP project are based on a combination of several modelling approaches (ocean biogeochemistry models combined with ocean circulation models, oceanic and atmospheric inversion models) while reported estimates in Zscheischler et al. (2017) are based on a combination of data-driven approaches (Landschützer et al., 2014; Rödenbeck et al., 2014). This comparison is interesting as it permits us

to compare the uncertainties introduced by the wind product alone to those resulting from a combination of different modeling approaches and different $pCO_2$ climatologies from observational data.

Figure 5 shows that most of our values fall within the range of values reported by RECCAP for all regions but the Southern Ocean, where the sink is clearly less intense. In this region, our results are more in line with those of Zscheischler et al. (2017), which can be explained by the fact that both studies rely on $pCO_2$ climatologies derived from observations and have performed a recalibration of the coefficient $c$ to fit the same mean global $k$ of 16 cm h$^{-1}$ (using solely the ERA wind product in Zscheischler et al., 2017). A likely explanation for the significant discrepancies with the RECCAP estimates is the weak observational constraints in this region (Landschützer et al., 2014; Rödenbeck et al., 2014). In other RECCAP regions, some of our $FCO_2$ estimates diverge from the range reported by Zscheischler et al. (2017, e.g. regions 1, 3 and 4). Likely explanations for these discrepancies are the inclusion of the Rödenbeck $pCO_2$ climatology in the study of Zscheischler et al. (2017) and the different periods of analysis (1991-2011 in our case $vs$ 2001-2010 in Zscheischler et al., 2017). Note also that we display all our data points while the uncertainties from both the RECCAP project and Zscheischler et al. (2017) are expressed as median absolute deviations (MAD, which correspond to the median of the absolute deviations from the data's median). Overall, this comparison reveals that the $FCO_2$ uncertainties associated with the choice of the $pCO_2$ climatology and its associated methodology calculated by Zscheischler et al. (2017) are larger than those associated to the choice of the wind product calculated in our study. The uncertainties reported by RECCAP which correspond to a compilation of various modelling approaches are even larger.

The differences between our different $FCO_2$ values calculated with $c*$ and compiled in Table 3, can be compared to the current estimates in $c$ value uncertainty. Over the years, this uncertainty decreased from about 30 % according to Sweeney et al. (2007) to about 10 % according to Ho et al. (2011). However, other sources of uncertainties are associated to $FCO_2$ in such a way that its cumulative value could reach 20 % (Wanninkhof, 2014). These additional sources of uncertainty are mainly attributed to the quantification of the Schmidt number (Jähne et al., 1987) and to estimation of $<U_{10}^2>$, especially at low (i.e. < 3 m s$^{-1}$) and high wind speed (i.e. > 12 m s$^{-1}$). At the global scale, the magnitude of the differences between the $FCO_2$ obtained using various combination of quadratics formulation of $k$ and wind products are similar to the range of uncertainty reported by Ho et al. (2011). However, the nature of the uncertainties reported in this study, which result from the wind speed products, is fundamentally different from those reported by Ho et al. (2011) which focuses on the experimental quantification of $c$. Moreover, the influence of changes in spatial and temporal resolution of the wind products should not be neglected as evidenced by the work by Naegler et al. (2006). This study indicates that a change in spatial resolution of the wind data from 4° x 5° degrees to 1° x 1° using monthly winds leads to discrepancies in $c$ values of about 3 % while the change in temporal resolution from daily to monthly using a 1° x 1° spatial resolution also leads to an uncertainty of about 3 %. Furthermore,Finally, as already pointed by Wanninkhof (1992), the use of monthly averaged values of $U_{10}$ instead of the 6 hour $<U_{10}^2>$ has a much bigger effect on $FCO_2$, with an underestimation reaching over 20 %.

It is also interesting to compare our reported $FCO_2$ uncertainties to those introduced by the choice of a given $pCO_2$ product. The application of distinct interpolation techniques in recent years has led to the publication of several global $pCO_2$ products that are largely based on the same observational dataset (i.e. SOCAT, Baker et al., 2016). To quantify the uncertainty introduced by the choice of the $pCO_2$ field, Rödenbeck et al. (2015) applied an identical parameterization of the $CO_2$ exchange at the air-water interface to 14 $pCO_2$ data products. The global $FCO_2$ ranged from -1.36 Pg C yr$^{-1}$ to -1.96 Pg C yr$^{-1}$, and the relative difference ($\sim$ 30 %) is thus slightly larger than the one attributed to different formulations of $k$ and wind products (20 %, ignoring NCEP2) calculated here.

## 5 Conclusions

Our study reinforces the notion that particular attention must be paid to the choice of the $k$-relationship and the wind product when calculating regional and global $FCO_2$ budgets. For global-scale applications, the most reliable approach to limit potential biases consists of fitting the coefficient $c$ for each wind product to match the globally average value of $k$ derived from $^{14}$C inventories ($c*$). Using this approach, we have shown that the uncertainty in $FCO_2$ attributable to the choice of the wind product is limited to about 10 % globally. Regionally, the uncertainties using $c*$ are significantly higher in the Equatorial Pacific (17 %) and in the Southern Ocean (14 %). Whenever the recalculation of $c$ is not possible, the choice of a formulation from the literature should be limited to the few recent formulations of $k$ derived from the global $^{14}$C inventory (i.e. Sweeney et al., 2007; Takahashi et al., 2009; Wanninkhof, 2014), as locally calibrated formulations of $k$ cannot be extrapolated globally and may yield widely different $FCO_2$ (up to $\sim$ 70 %) when all wind products and k formulations are included. In addition, even in this case, we recommend favoring the use of the wind speed product that was originally applied to derive the value of $c$ (Naegler et al., 2006), further noting that, a change in the spatial or temporal resolution at which calculations are performed may yield additional uncertainties (Naegler et al., 2006; Wanninkhof, 2014). Our calculations reveal that, whenever a formulation of $k$ is used to quantify the global oceanic $FCO_2$ indistinctly with ERA, CCMP or NCEP1, the range of estimates will be associated with an uncertainty of the order of 12 % when combined with recent global formulation of $k$ derived from the $^{14}$C global inventory, only. This uncertainty significantly rises when using the out dated formulation proposed by Wanninkhof (1992), a hybrid $k$-formulation (Wanninkhof et al., 2009) and/or when $FCO_2$ is calculated with NCEP2. Furthermore, our results have highlighted that due to differences in the regional wind patterns, regional discrepancies in $FCO_2$ are even larger than global. Finally, other poorly constrained sources of uncertainty in the calculation of $FCO_2$ and not included in our study exist in polar and coastal regions when specific processes further complicate the air-water exchange. For instance, in partially ice covered areas, the relationship between the intensity of the gas exchange is more complex than a direct linear scaling to the ice-free surface area (Lovely et al., 2015), but no generic formulation exists yet to account for this effect. Similarly, in some coastal areas, specific physical processes such as the occurrence of surfactants or other sources of turbulences than wind such as tidal currents may affect the intensity of the exchange of $CO_2$ at the air-water interface (Ho et al., 2011). In the future, the quantification of the effect of such processes

on the uncertainty over the air-water $CO_2$ exchange will have to be further investigated to better constrain regional carbon budgets. It should be noted that it is difficult to directly extrapolate our results to $FCO_2$ derived from global circulation models and Earth System Models. Indeed, because of the dynamic air-sea $pCO_2$ gradient adjustment acting against the change in gas transfer velocity in these models, the effect of variations in $k$ on global $FCO_2$ estimates are dampened. For instance, Sarmiento et al. (1992) showed that a doubling in $k$ resulted in only about a 10 % increase in the overall anthropogenic $CO_2$ absorption by the ocean. Because of the absence of this negative feedback mechanism in observation-based estimates, it is expected that wind-induced uncertainties derived from observations will be larger than uncertainties derived from Ocean general circulation models (OGCMs) and Earth System Models. Furthermore, the use of a linear $k$ formulation and a single wind product in Sarmiento et al. (1992) will lead to smaller uncertainties than in our assessment based on quadratic formulations and multiple wind products. As shown by the results of Ishii et al. (2014) for the Pacific Ocean, significant $FCO_2$ differences can be observed using the same model but different wind products. Currently, the uncertainty of the global $CO_2$ uptake by the ocean is estimated by comparing multiple global models (Ciais et al., 2013). Unfortunately, these models use various formulations of $k$, some of which outdated like that of Wanninkhof (1992) and wind products which are not always consistent with their formulation of the piston velocity. Based on our analysis of the impact of the choice of the wind product and its resolution on $FCO_2$, we believe it would be beneficial to update these representations of the $CO_2$ exchange in these models. Ideally, the value of $c$ should be adapted to match a global $k$ consistent with the global average derived from the latest [14]C budget (Wanninkhof et al., 2014).

**Data availability**

The observation-based global monthly gridded sea surface $pCO_2$ product is provided by Landschützer, P., N. Gruber and D.C.E. Bakker. 2015: "A 30 years observation-based global monthly gridded sea surface pCO2 product from 1982 through 2011". http://cdiac.ornl.gov/ftp/oceans/SPCO2_1982_2011_ETH_SOM_FFN. Carbon Dioxide Information Analysis Center, Oak Ridge National Laboratory, US Department of Energy, Oak Ridge, Tennessee. doi: 10.3334/CDIAC/OTG.SPCO2_1982_2011_ETH_SOM-FFN. The global atmospheric reanalysis ERA-interim datasets (ERA, Dee et al., 2011, http://doi.wiley.com/10.1002/qj.828) are accessible on the European Centre for Medium-Range Weather Forecasts (ECMWF) web site. The Cross-Calibrated Multi-Platform Ocean Wind Vector 3.0 datasets (CCMP, Atlas et al., 2011, doi:10.1175/2010BAMS2946.1) are provided by the NASA/GSFC/NOAA. 2009. Cross-Calibrated Multi-Platform Ocean Surface Wind Vector L3.0 First-Look Analyses. Ver. 1. PO.DAAC, CA, USA. http://dx.doi.org/10.5067/CCF30-01XXX. The NCEP/NCAR reanalysis 1 (NCEP1, Kalnay et al., 1996, doi:10.1175/1520-0477(1996)077<0437:TNYRP>2.0.CO;2) and the NCEP/DOE AMIP-II Reanalysis (NCEP2, Kanamitsu et al., 2002, doi:10.1175/BAMS-83-11-1631) provided by the NOAA/OAR/ESRL PSD, Boulder, Colorado, USA, from their Web site at http://www.esrl.noaa.gov/psd/data/gridded/. The NOAA High Resolution SST data is provided by the NOAA/OAR/ESRL PSD, Boulder, Colorado, USA, from their Web site at http://www.esrl.noaa.gov/psd/.

## Competing interests

The authors declare that they have no conflict of interest.

## Acknowledgements

G. G. Laruelle is postdoctoral researcher of F.R.S.-FNRS at the Université Libre de Bruxelles. P. Landschützer is supported by the Max Planck Society for the Advancement of Science. The research leading to these results has received funding from the European Union's Horizon 2020 research and innovation programs under the Marie Sklodowska-Currie grant agreement no. 643052 (C-CASCADES project) and under a grant agreement No 776810 (VERIFY project).

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

**Table 1: Wind products used in this study.** $U_{10}$ **represent the wind speed measured 10 meters above sea level.** $\overline{<U_{10}>}$ **and** $\overline{<U_{10}^2>}$ **represent the global 21 year monthly mean and the second moment on a centered 1° x 1° spatial resolution grid, respectively.**

| | CCMP | ERA | NCEP1 | NCEP2 |
|---|---|---|---|---|
| Name | « Cross Calibrated Multi Plateform » | « ERA-Interim » | « NCEP/NCAR reanalysis 1 » | « NCEP/DOE reanalysis 2 » |
| Temporal range | 1991-2011 | 1991-2011 | 1991-2011 | 1991-2011 |
| Temporal resolution | 6 hours | 6 hours | 6 hours | 6 hours |
| Spatial resolution | 0.25° x 0.25° | 0.25° x 0.25° | T62 Gaussian | T62 Gaussian |
| $\overline{<U_{10}>}$ (m s$^{-1}$) | 7.55 | 7.36 | 7.20 | 8.21 |
| $\overline{<U_{10}^2>}$ (m$^2$ s$^{-2}$) | 69.29 | 66.36 | 64.04 | 85.50 |

**Table 2: Historical summary of $k$-relationships using $U_{10}$ in the ocean. Depending on the study, $k$ is either expressed as $k_{660}$ or $k_{600}$, which represent the gas exchange transfer velocity of $CO_2$ at 20°C in seawater ($SSS = 35$) and freshwater ($SSS = 0$), respectively. Also note that some equations are developed for the use of monthly mean wind speeds (denoted "long term") while others are developed for instantaneous daily/weekly wind speed (see Wanninkhof (1992) for details). The equations used in our analysis are identified in bold.**

| Study | Campaign and location | Methodology | k-parametrization |
|---|---|---|---|
| **Wanninkhof (1992)** | - | based on a Rayleigh distribution of $U_{10}$, a mean gas invasion rate of 21 cm h$^{-1}$ to fit the global ocean bomb $^{14}$C inventory estimated by Broecker et al. (1985) | $\mathbf{k_{660}= 0.31u_{10}{}^2}$ <br> $k_{660}= 0.39u_{10}{}^2$ (long term wind) |
| Wanninkhof and McGillis (1999) | "Gas Ex-98" North Atlantic | covariance flux and air-water $\Delta pCO_2$ disequilibrium results and constrained to the global ocean bomb $^{14}$C inventory from Broecker et al. (1985) | $k_{660}= 0.0283u_{10}{}^3$ <br> $k_{660} = 1.09U_{10}-0.333u_{10}{}^2+0.078u_{10}{}^3$ (long term wind) |
| Jacobs et al. (1999) | "ASGAMAGE" Dutch coast | covariance flux and air-water concentration difference results | $k_{660} = 0.54u_{10}{}^2$ |
| Nightingale et al. (2000) | Southern North Sea | compilation of dual-deliberate tracers results | $k_{600} = 0.222u_{10}{}^2+0.333u_{10}$ |
| McGillis et al. (2001) | "Gas Ex-98" North Atlantic | covariance flux, dual-deliberate tracers, atmospheric $CO_2$ and dimethylsulfide profiles, and water column mass balance for $CO_2$ results | $k_{660} = 3.3+0.026U_{10}{}^3$ |
| Kuss et al. (2004) | Eastern Gotland Sea | surface water total $CO_2$ concentration results | $k_{660} = 0.45u_{10}{}^2$ <br> $k_{660} = 0.037u_{10}{}^3$ |
| McGillis et al. (2004) | "Gas Ex-2001" Equatorial Pacific | covariance flux and flux profile results | $k_{660} = 8.2+0.014u_{10}{}^3$ |
| Ho et al. (2006) | "SAGE" Western Pacific sector of the Southern Ocean | compilation of dual-deliberate tracers results and constrained with the new estimate of global ocean excess $^{14}$C uptake from Naegler et al. (2006) | $k_{600} = 0.266u_{10}{}^2$ |

| | | | |
|---|---|---|---|
| **Sweeney et al. (2007)** | - | global approach using ocean general circulation models in an inverse mode, a new ocean bomb $^{14}$C inventory and the second moment of the 6 hours 3.75° x 4.5° NCEP1 wind speed product (Kalnay et al., 1996) | $\mathbf{k_{660} = 0.27u_{10}{}^2}$ |
| Weiss et al. (2007) | Southern Baltic Sea | covariance flux results | $k_{660} = 0.365u_{10}{}^2 + 0.46u_{10}$ |
| **Takahashi et al. (2009)** | - | global approach using ocean general circulation models, the global ocean bomb $^{14}$C inventory from Sweeney et al. (2007) and the second moment of the 6 hours 4° x 5° NCEP2 wind speed data base (Kanamitsu et al., 2002) | $\mathbf{k_{660} = 0.26u_{10}{}^2}$ |
| **Wanninkhof et al. (2009)** | - | based on a conceptual model that incorporates processes which affect gas transfer velocity. Coefficients are calculated based on information from the literature | $\mathbf{k_{660} = 3 + 0.1u_{10} + 0.064u_{10}{}^2 + 0.011u_{10}{}^3}$ $k_{660} = 0.24u_{10}{}^2$ |
| Prytherch et al. (2010) | "UK-SOLAS project HiWaSE" North Atlantic | covariance flux results | $k_{660} = 5.3 + 0.034u_{10}{}^3$ |
| Ho et al. (2011) | "SO Gas Ex " Southwest Atlantic | compilation of dual-deliberate tracers results | $k_{600} = 0.262u_{10}{}^2$ |
| Edson et al. (2011) | "SO Gas Ex" Southern Atlantic | covariance flux results combined with data from Gas Ex-98 and Gas Ex-2001 | $k_{660} = 5.4 + 0.029u_{10}{}^3$ |
| **Wanninkhof (2014)** | - | global approach using an ocean inverse model, the global ocean bomb $^{14}$C inventory from Sweeney et al. (2007) and the second moment of the 6 hours 0.25° x 0.25° CCMP wind speed product (Atlas et al., 2011) | $\mathbf{k_{660} = 0.251u_{10}{}^2}$ |

**Table 3: Relative difference in $FCO_2$ between the highest and lowest $FCO_2$ obtained using different combinations of wind products and $k$-formulations expressed in percent. Results are calculated globally (numbers in bold) and regionally (minimum and maximum relative difference between all RECCAP regions results). Note that the RECCAP regions 2 and 3 (Equatorial Pacific, EP) have been merged. $C^*$ represents $FCO_2$ calculated with a quadratic $k$-relationship where $c$ is recalibrated for each wind product to fit a global average $k$ value of 16 cm h$^{-1}$ for the period of our study (1991-2011).**

| | One wind product | | 3 wind products (CCMP, ERA, NCEP1) | | All wind products | |
|---|---|---|---|---|---|---|
| One quadratic | **-** | - | **6** | < 10 except EP | **29** | 21-30 |
| Quadratics | **7 (19[a])** | 7 (19[a]) | **12 (24[a])** | 11-29 (22-38[a]) | **34 (43[a])** | 27-35 (36-43[a]) |
| Quadratics[b] + hybrid | **~ 13** | 7-16 | **20** | 13-29 | **40** | 27-40 |
| C* | **-** | - | **10** | 3-17 | **11** | 3-17 |

a including results calculated with the k-relationship of Wanninkhof et al. (1992)
b excluding results calculated with the k-relationship of Wanninkhof et al. (1992)

**Table 4: Global 21 year mean oceanic air-sea $FCO_2$ (Pg C yr$^{-1}$) calculated using several $k$-relationships and $c*$ combined with the different wind speed products. Numbers in bold represent the minimum and maximum $FCO_2$ values obtained out of all the possible combinations.**

| | FCO2-CCMP | FCO2-ERA | FCO2-NCEP1 | FCO2-NCEP2 |
|---|---|---|---|---|
| | (Pg C yr$^{-1}$) | | | |
| Global k-relationships | | | | |
| | | | | |
| Wanninkhof (1992) | -1.67 | -1.60 | -1.70 | -2.27 |
| Sweeney et al. (2007) | -1.46 | -1.40 | -1.48 | -1.98 |
| Takahashi et al. (2009) | -1.40 | -1.35 | -1.43 | -1.90 |
| Wanninkhof (2014) | -1.35 | -1.30 | -1.38 | -1.84 |
| C* | -1.38 | -1.40 | -1.53 | -1.54 |
| Wanninkhof et al. (2009) | -1.27 | **-1.19** | -1.25 | -1.86 |
| Empirical k-relationships | | | | |
| Kuss et al. (2004) (quadratic) | -2.43 | -2.33 | -2.47 | -3.29 |
| Kuss et al. (2004) (cubic) | -2.68 | -2.48 | -2.61 | **-4.23** |
| Weiss et al. (2007) | -2.15 | -2.07 | -2.20 | -2.89 |
| Prytherch et al. (2010) | -2.61 | -2.42 | -4.04 | -2.54 |
| Edson et al. (2011) | -2.25 | -2.09 | -3.47 | -2.19 |
| Ho et al. (2011) | -1.35 | -1.29 | -1.37 | -1.83 |

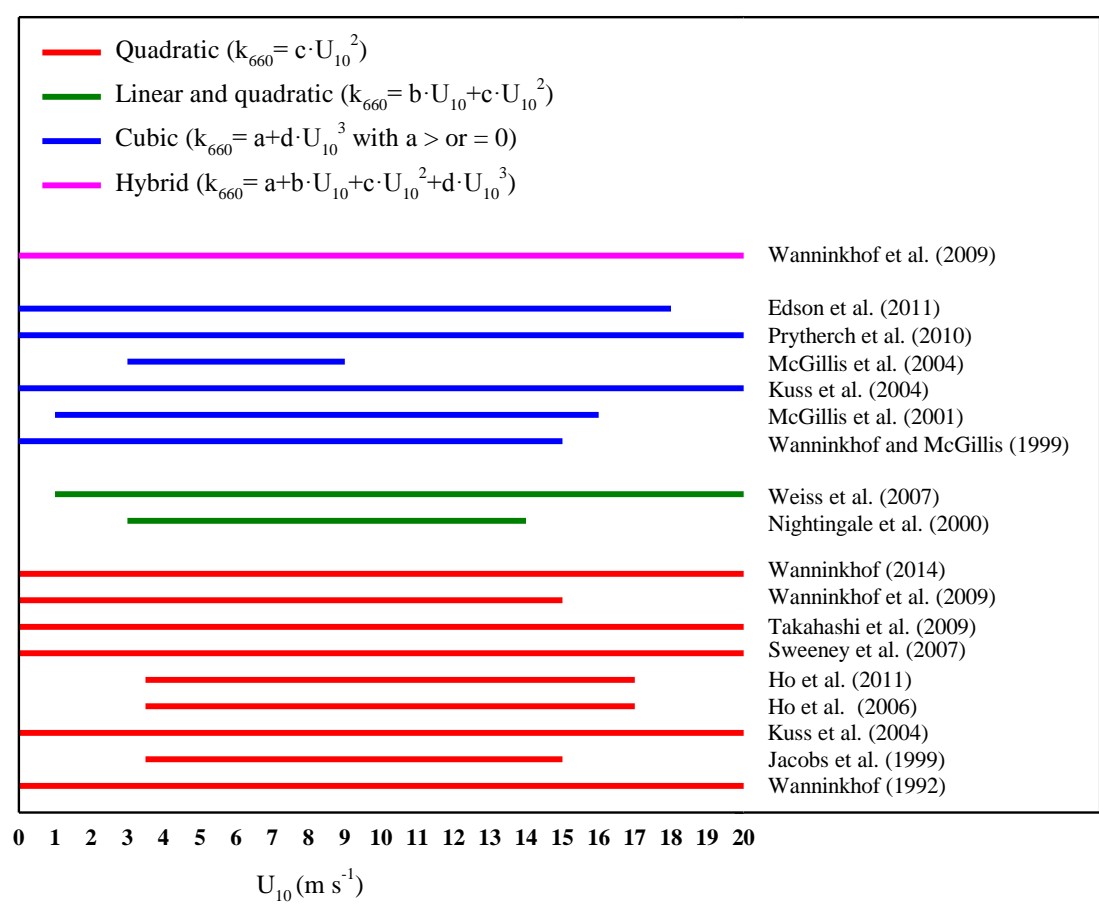

**Figure 1: List of *k-U₁₀* relationships (quadratic, cubic, linear and quadratic and hybrid) reported in the literature for the ocean with their range of applicability.**

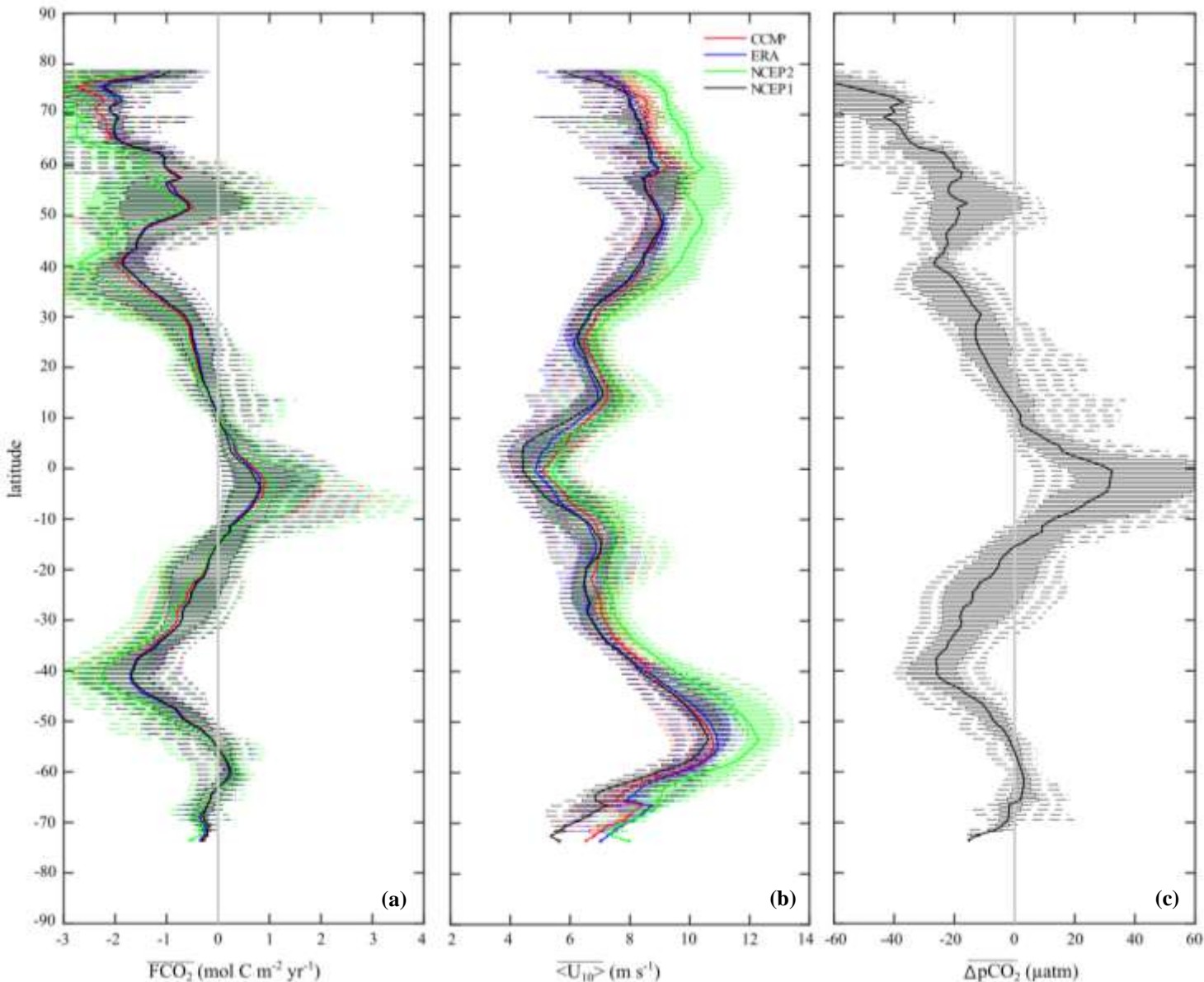

**Figure 2: Latitudinal distribution of** $FCO_2$ **(mol C m$^{-2}$ yr$^{-1}$) (a),** $U_{10}$ **(m s$^{-1}$) (b) and** $\Delta pCO_2$ **(μatm) (c) using the CCMP, ERA, NCEP1 and NCEP2 wind products.** $FCO_2$ **is calculated using the quadratic** $k$-$U_{10}$ **relationship from Wanninkhof (2014). Results refer to the 1991-2011 period. The median value for each latitude is represented by a line, while the box plots delineate the 5$^{th}$ and 95$^{th}$ percentile of the variation within each 1° latitudinal band, respectively.**

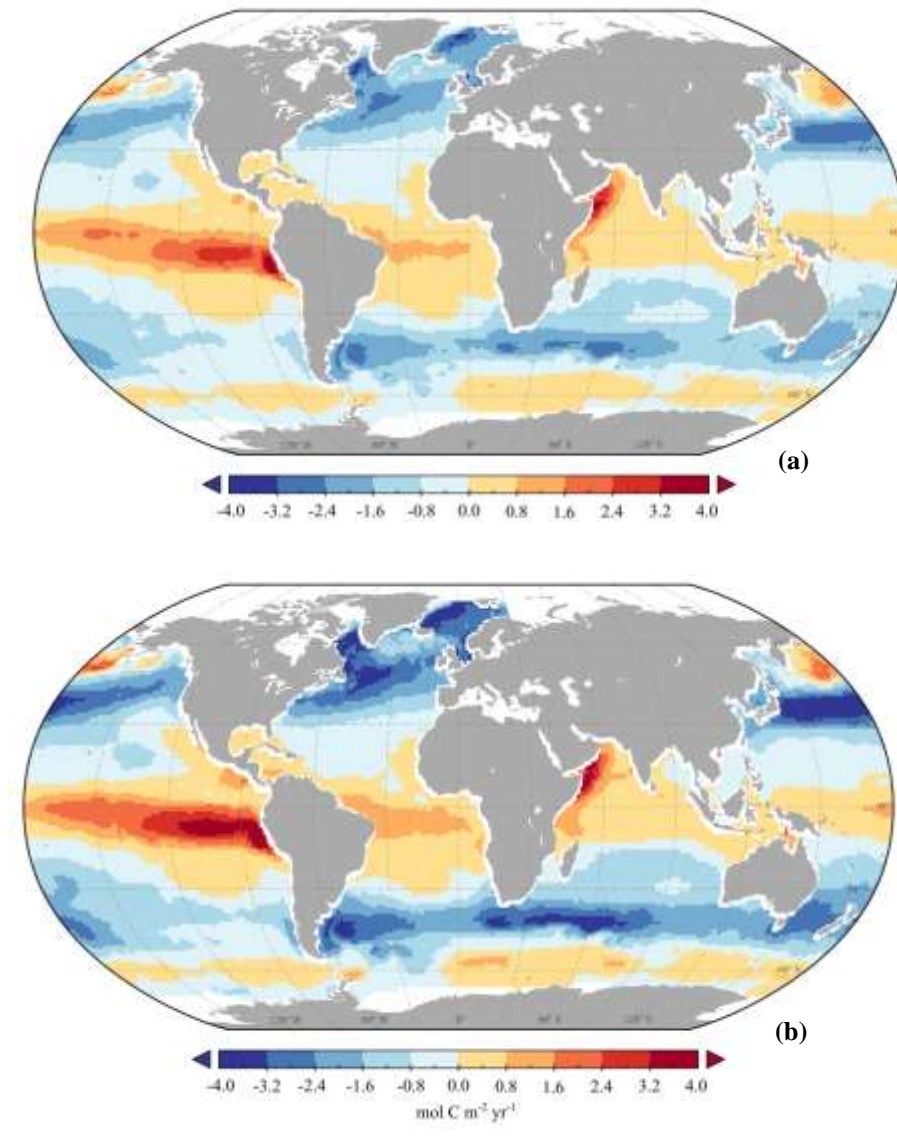

**Figure 3: Global distributions of oceanic air-sea mean $FCO_2$ (mol C m$^{-2}$ yr$^{-1}$) generated from a 21 year climatology (1991-2011) using the Wanninkhof (2014) *k*-relationship combined with NCEP1 (a) and NCEP2 (b) wind products.**

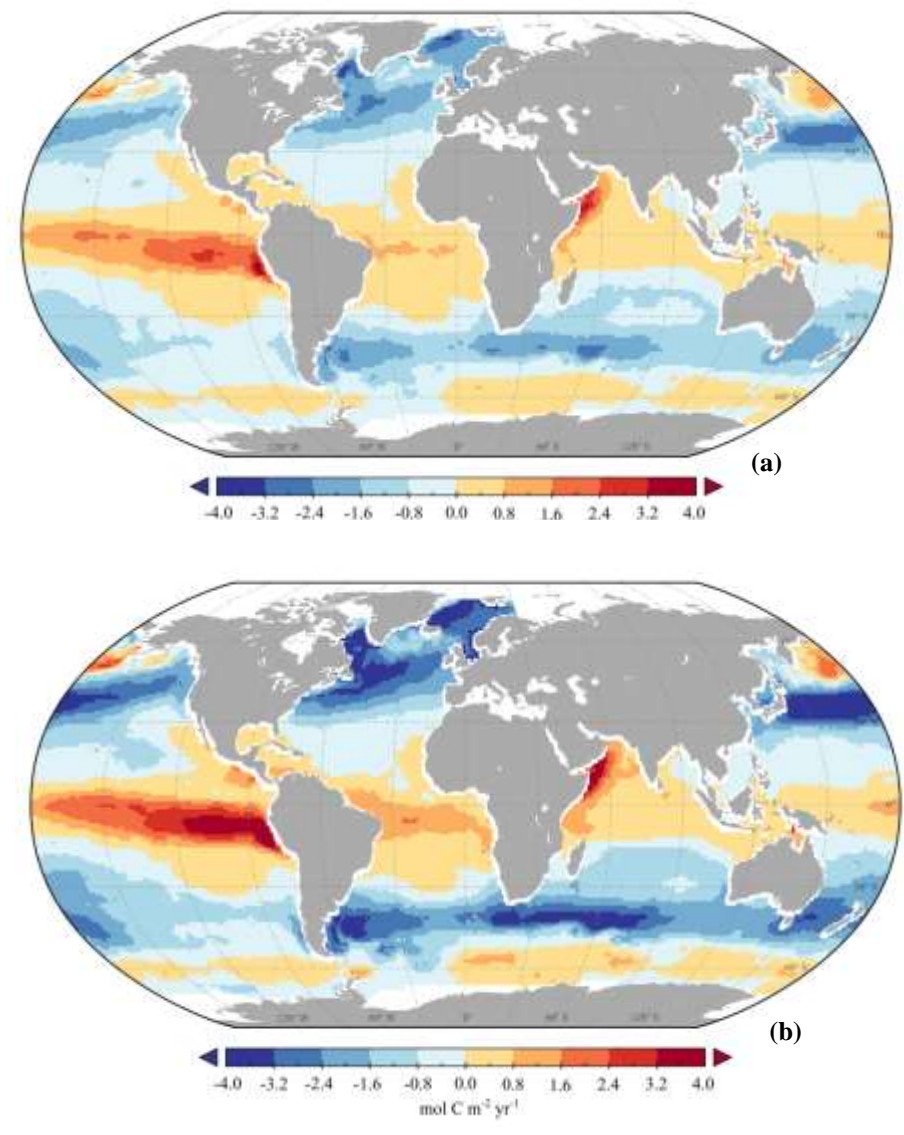

**Figure 4: Global distributions of oceanic air-sea mean $FCO_2$ (mol C m$^{-2}$ yr$^{-1}$) generated from a 21 year climatology (1991-2011) using the Wanninkhof et al. (2009) $k$-parametrization combined with NCEP1 (a) and the Sweeney et al. (2007) $k$-relationship combined with NCEP2 (b)**

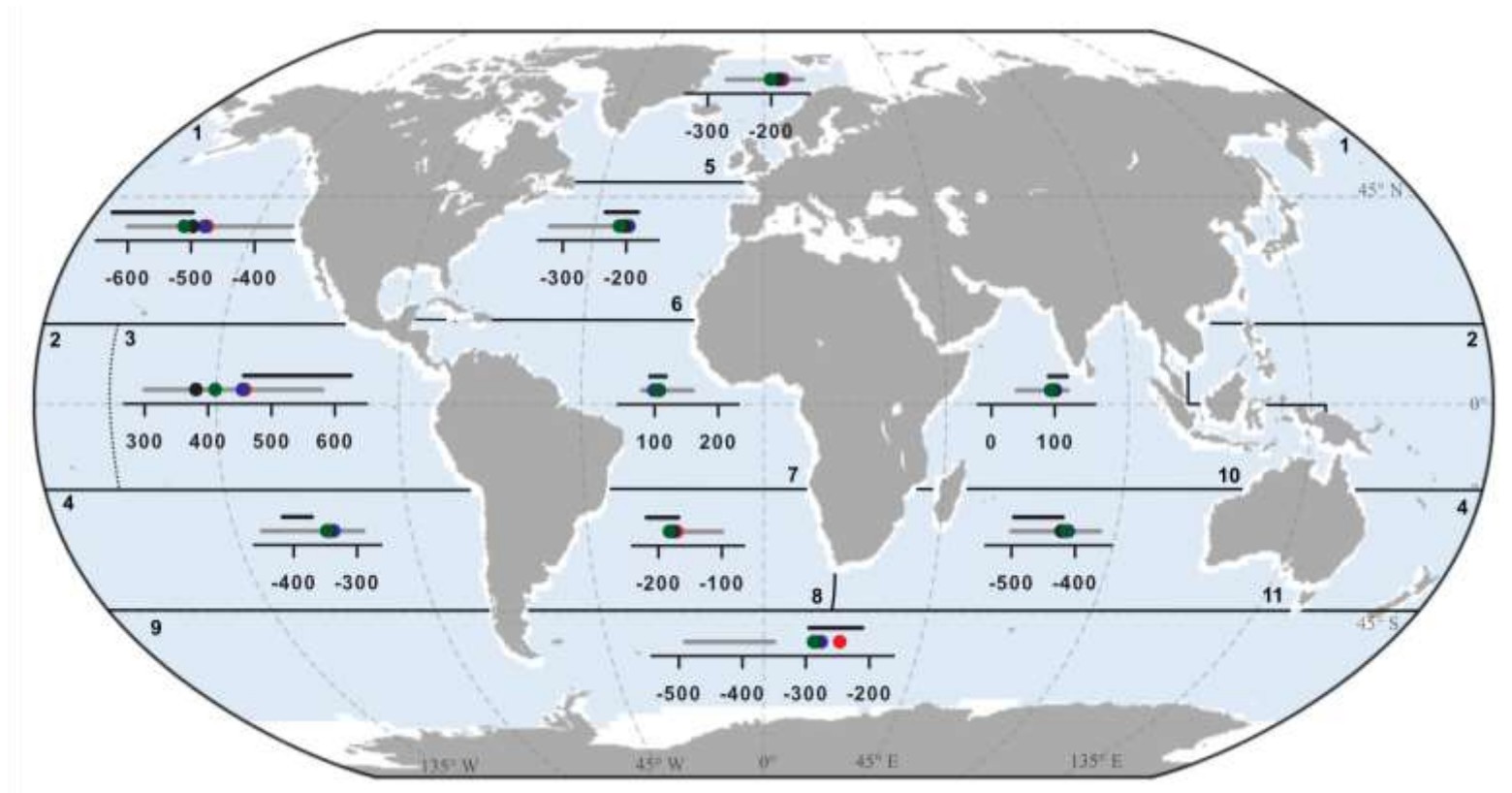

Figure 5: Spatial extent of the 11 RECCAP regions used for our regional analysis. In each regions the range of uncertainties in $FCO_2$ (Tg C yr$^{-1}$) is represented as calculated by the different RECCAP studies (grey lines; Ishii et al., 2014; Lenton et al., 2013; Sarma et al., 2013; Schuster et al., 2013) and by Zscheischler et al. (2017, black lines) for the 1990-2009 and 2001-2010 periods, respectively. These ranges are represented by the median absolute deviation (MAD). Points correspond to the $FCO_2$ (Tg C yr$^{-1}$) calculated in our study with a quadratic k-relationship where $c$ is recalibrated for each wind product to fit a global average $k$ value of 16 cm h$^{-1}$ for the 1991-2011 period ($c^*$). $FCO_2$ points calculated with CCMP, ERA, NCEP1 and NCEP2 are represented in red, blue, black and green respectively. In region 5, the $FCO_2$ range calculated by Zscheischler et al. (2017) is not represented because they take into account the Arctic region which is not include in our study.

**Table S1:** 21 year mean $U_{10}$ ($\overline{\langle U_{10} \rangle}$, m s$^{-1}$) and $FCO_2$ ($\overline{FCO_2}$, Tg C yr$^{-1}$) in the 11 oceanic RECCAP regions calculated using the different global $k$-relationships combined with four different wind products. C* refers to $FCO_2$ calculated with a quadratic $k$-relationship where $c$ is recalibrated for each wind product to fit a global average $k$ value of 16 cm h$^{-1}$ for the period of the study (1991-2011). Following the RECCAP nomenclature, NP stands for North Pacific, EP: Equatorial Pacific, SP: South Pacific, NA: North Atlantic, EA: Equatorial Atlantic, SA: South Atlantic, SO: Southern Ocean, NI: North Indian Ocean and SI: South Indian Ocean. Results using the $k$-parametrization of Wanninkhof (1992), which are excluded from our analysis, are represented in italic. The interannual variability is reported as standard deviations between brackets.

| regions numbers | NP 1 | EP 2 | EP 3 | SP 4 | NA 5 | NA 6 | EA 7 | SA 8 | SO 9 | NI 10 | SI 11 |
|---|---|---|---|---|---|---|---|---|---|---|---|
| $\overline{\langle U_{10} \rangle}$ (m s$^{-1}$) | | | | | | | | | | | |
| CCMP | 7.6 (0.2) | 6.1 (0.3) | 6.5 (0.3) | 7.4 (0.2) | 9.0 (0.3) | 7.3 (0.2) | 6.5 (0.2) | 7.6 (0.2) | 9.84 (0.3) | 6.1 (0.2) | 8.1 (0.2) |
| ERA | 7.4 (0.1) | 5.8 (0.2) | 6.2 (0.3) | 7.1 (0.1) | 8.8 (0.2) | 7.0 (0.1) | 6.2 (0.1) | 7.4 (0.1) | 10.0 (0.2) | 5.9 (0.1) | 7.8 (0.1) |
| NCEP1 | 7.4 (0.1) | 5.6 (0.1) | 5.7 (0.2) | 7.0 (0.2) | 8.7 (0.2) | 7.0 (0.1) | 6.4 (0.1) | 7.4 (0.1) | 9.5 (0.2) | 5.8 (0.1) | 7.8 (0.1) |
| NCEP2 | 8.3 (0.1) | 6.2 (0.1) | 6.8 (0.3) | 7.8 (0.2) | 10.2 (0.2) | 7.8 (0.1) | 7.1 (0.1) | 8.2 (0.1) | 11.1 (0.3) | 6.6 (0.1) | 8.7 (0.1) |
| $\overline{FCO_2}$ (Tg C yr$^{-1}$) | | | | | | | | | | | |
| **CCMP** | | | | | | | | | | | |
| Sweeney et al. (2007) | -500 (106) | 36 (32) | 447 (74) | -365 (73) | -191 (35) | -211 (37) | 109 (34) | -181 (36) | -260 (152) | 100 (29) | -442 (37) |
| Takahashi et al. (2009) | -482 (102) | 35 (31) | 430 (71) | -351 (71) | -184 (34) | -203 (36) | 105 (32) | -174 (35) | -250 (146) | 97 (28) | -425 (35) |
| Wanninkhof (2014) | -465 (99) | 34 (30) | 415 (69) | -339 (68) | -178 (33) | -196 (35) | 102 (31) | -168 (34) | -242 (141) | 93 (27) | -411 (34) |
| Wanninkhof et al. (2009) | -439 (94) | 36 (27) | 381 (59) | -312 (63) | -172 (32) | -186 (33) | 94 (28) | -157 (32) | -222 (135) | 93 (25) | -381 (32) |
| C* | -474 (101) | 35 (30) | 424 (70) | -346 (70) | -181 (33) | -200 (35) | 104 (32) | -171 (34) | -246 (144) | 95 (27) | -419 (35) |
| *Wanninkhof (1992)* | *-574 (122)* | *42 (37)* | *513 ( 85)* | *-419 (84)* | *-220 (40)* | *-242 (43)* | *125 (38)* | *-207 (41)* | *-298 (174)* | *115 (33)* | *-507 (42)* |
| **ERA** | | | | | | | | | | | |
| Sweeney et al. (2007) | -477 (91) | 35 (30) | 418 (70) | -336 (59) | -184 (28) | -195 (32) | 98 (31) | -175 (33) | -275 (156) | 101 (28) | -409 (22) |
| Takahashi et al. (2009) | -459 (87) | 33 (29) | 402 (67) | -323 (56) | -177 (27) | -188 (31) | 95 (30) | -169 (31) | -264 (150) | 97 (27) | -394 (22) |
| Wanninkhof (2014) | -443 (84) | 32 (28) | 389 (65) | -312 (54) | -171 (26) | -181 (30) | 92 (29) | -163 (30) | -255 (145) | 94 (26) | -380 (21) |
| Wanninkhof et al. (2009) | -413 (79) | 35 (26) | 360 (56) | -286 (50) | -162 (25) | -169 (28) | 87 (26) | -151 (29) | -236 (138) | 94 (25) | -351 (19) |
| C* | -479 (91) | 35 (30) | 420 (70) | -337 (59) | -184 (28) | -196 (32) | 99 (32) | -176 (33) | -276 (156) | 101 (28) | -410 (22) |
| *Wanninkhof (1992)* | *-548 (104)* | *40 (35)* | *480 (80)* | *-385 (67)* | *-211 (32)* | *-224 (37)* | *113 ( 36)* | *-201 (38)* | *-315 (179)* | *116 (32)* | *-470 (26)* |

**NCEP1**

| | | | | | | | | | | | |
|---|---|---|---|---|---|---|---|---|---|---|---|
| Sweeney et al. (2007) | -481 (88) | 21 (26) | 347 (56) | -334 (60) | -184 (28) | -197 (32) | 104 (33) | -173 (33) | -276 (139) | 95 (29) | -408 (19) |
| Takahashi et al. (2009) | -463 (85) | 21 (25) | 335 (54) | -322 (57) | -177 (27) | -189 (31) | 100 (32) | -167 (32) | -265 (134) | 91 (28) | -393 (18) |
| Wanninkhof (2014) | -447 (82) | 20 (24) | 323 (52) | -311 (55) | -171 (26) | -183 (30) | 96 (31) | -161 (31) | -256 (129) | 88 (27) | -379 (18) |
| Wanninkhof et al. (2009) | -418 (77) | 27 (23) | 316 (47) | -285 (51) | -163 (25) | -171 (28) | 91 (28) | -149 (29) | -239 (123) | 89 (25) | -349 (16) |
| C* | -497 (91) | 22 (27) | 359 (58) | -345 (62) | -190 (29) | -203 (33) | 107 (34) | -179 (34) | -285 (144) | 98 (30) | -422 (20) |
| *Wanninkhof (1992)* | *-552 (101)* | *24 (30)* | *399 (64)* | *-384 (68)* | *-211 (32)* | *-226 (37)* | *119 (38)* | *-199 (38)* | *-316 (160)* | *109 (33)* | *-468 (22)* |

**NCEP2**

| | | | | | | | | | | | |
|---|---|---|---|---|---|---|---|---|---|---|---|
| Sweeney et al. (2007) | -654 (121) | 31 (33) | 495 (74) | -445 (78) | -257 (41) | -269 (44) | 135 (42) | -234 (44) | -368 (197) | 120 (37) | -530 (23) |
| Takahashi et al. (2009) | -629 (117) | 30 (32) | 477 (71) | -429 (75) | -248 (39) | -259 (42) | 130 (40) | -225 (42) | -354 (190) | 115 (36) | -510 (22) |
| Wanninkhof (2014) | -608 (113) | 29 (31) | 460 (69) | -414 (73) | -239 (38) | -250 (41) | 125 (39) | -217 (41) | -342 (183) | 111 (35) | -492 (21) |
| Wanninkhof et al. (2009) | -604 (112) | 32 (28) | 419 (61) | -397 (70) | -245 (40) | -253 (41) | 113 (34) | -215 (41) | -340 (188) | 108 (32) | -474 (20) |
| C* | -511 (95) | 24 (26) | 387 (58) | -348 ( 61) | -201 (32) | -211 (34) | 105 (33) | -183 (34) | -288 (154) | 93 (29) | -414 (18) |
| *Wanninkhof (1992)* | *-750 (139)* | *36 (38)* | *569 ( 85)* | *-511 ( 90)* | *-295 (47)* | *-309 (50)* | *155 (48)* | *-268 (50)* | *-423 (227)* | *137 (43)* | *-608 (26)* |