# Peer review of "Uncertainty of the global oceanic $CO_2$ uptake induced by wind forcing: quantification and spatial analysis"

_Biogeosciences, 2017_

## Referee Comment (RC1) · Anonymous Referee #1 · 7 Dec 2017

The authors have conducted a very nice study that works through uncertainties inherent in the global uptake of CO2 associated with wind forcing uncertainties. The presentation is clear, the methods are transparent, and the results should be of broad interest to the ocean carbon cycle community. The only shortcoming of the study is that it is somewhat technical, and in order to satisfy the scientific priorities of Biogeosciences my recommendation would be that the manuscript would benefit from minor revisions before being accepted for publication. A number of more general and more specific questions/comments are raised in the text that follows.

Broad Comments:

First and foremost, in the discuss pertaining to Fig. 2 and Fig. 3 and Fig. 4, it would be useful if the authors could identify whether they authors see pertinent dynamical or circulation structures as dominating the uncertainties. Given the linear color scale in Fig. 3 and Fig. 4, it appears that the dominant uncertainties might be mostly over western boundary current regions? This would appear to be important, as there is currently a question being discussed in the carbon cycle community of whether western boundary currents serve as hot spots for carbon uptake. It would be of wide interest if the authors could attempt to quantify a response to the following question: Even with a perfect observing system for perfect $pCO_2$ measurements that is seasonally-resolving, what inherent uncertainties from gas exchange parameterizations and winds apply over these particular structures? This is already implicitly there in the text, but it would be helpful to emphasize this a bit to make clear what challenges lie ahead for more regionals-focused mechanistic interpretations.

It would also be beneficial for anchoring the present study in the published literature if the authors could relate their results to the study of Sarmiento, Orr, and Siegenthaler (1992; JGR), where it was reported within a modeling context that 100% increases in gas exchange only impact 9.2% changes in air-sea $CO_2$ fluxes. Although that study from 25 years ago used a simplified representation of anthropogenic carbon, it has long been cited for the argument that gas exchange representation isn't of critical importance for global uptake.

In a related point, it would also be beneficial if the authors could likewise relate the uncertainties here to those reported in the Rödenbeck et al. (2015) comparison of fluxes found for different gridded $pCO_2$ products from different global research groups.

The authors point out that there are important discrepancies between global and empirical formulations of gas exchange. It could be constructive in this regard to point out that there are expected to be important limitations with the construct of a piston velocity in representing the relationship between surface carbon fluxes and winds. Winds don't only impact air-sea fluxes through microturbulence at the air-sea interface, but

in the sense of climate dynamics the very same winds can in some regions sustain entrainment of waters through deepening of the mixed lawyer (through shear-induced turbulence). As the relative amplitude of these processes is not known, one should expect that parameterizations based on the concept of piston velocity (the first of these) will have limitations (Rodgers et al., 2014, BG), and that parameterizations based on local properties at the air-sea interface may thereby have inherent uncertainties that are irreducible. I think that this point in fact strengthens the main arguments in the manuscript, in that systematic and rigorous analysis of uncertainties will continue to be a critical component of carbon cycle research moving forward. Regarding the first of the points above (western boundary currents), this could be pertinent.

Minor Comments:

pg. 2, line 5 "open" should be "opened"

pg. 2, line 12: "observation-based" should probably be "observationally-based"

pg 2 line 19 should be "10 meters"

pg 2 line 28: should be "uncertainty associated with"

pg 3 lines 3-4 should be "the latitudinal distribution of FCO2"

pg 3 line 10 I recommend removing "in a nutshell" (rather informal)

pg 3 line 14 change "measure because z" to "measure as z"

pg 4 line 12 again, I think this should say "observationally-based"

pg 4 line 18 I think this should say "using a two-step"

pg 4 line 19 should say "maps for the global ocean"

pg 7 line 17 should say "Within the tropics"

pg 9 line 21 should say "This corresponds"

[Figure]

pg 9 line 22 should say "despite the fact that these empirical"

pg 11 line 16 replace "the literature also reports" with "one also finds in the published literature"

pg 11 line 24 should say "methods have their relative advantages"

pg 11 line 24 should say "has been shown that"

Summary Statement:

To restate, I believe that with some relatively minor text changes that connect the present study to broader community efforts and scientific interests, the manuscript should meet the standard for publication in Biogeosciences. The authors are to be commended for a very nice and thorough analysis and presentation that will be of brand interest.

---

## Referee Comment (RC2) · Anonymous Referee #2 · 15 Jan 2018

Review of manuscript "Uncertainty of the global oceanic CO2 uptake induced by wind forcing: quantification and spatial analysis" by Alizee Roobaert, Goulven G. Laruelle, Peter Landschützer, and Pierre Regnier

In their manuscript the authors present an in depth study of the impact of different gridded wind speed data products - such as CCMP, ERA, NCEP1 and NCEP2 - utilized to determine air sea fluxes of carbon dioxide on global and regional scales. Therefore they combined those data with a globally re-gridded sea surface climatology of pCO2 values covering the years from 1991 to 2011. By employing different parameterizations for the calculation of CO2 fluxes they found a strong dependence of this number on the

choice of the specific wind speed product.

To constrain the variability of air to sea fluxes of CO2 is of great importance for assessing the global ocean carbon sink and the concomitant acidification. In order to minimize the error in the flux calculations the authors propose to recalibrate the piston velocities (k-formulations) for the respective wind speed data product.

The manuscript is well written and contains important informations and innovations. The methods and results are clearly presented. However, the manuscript would benefit from a short discussion of the consequences for Earth system modeling. What is the expected impact of the findings in this study on model projections regarding the evolution of the future carbon sink? A short clarifying paragraph would be helpful.

I recommend publication in "Biogeosciences"

---

## Author Comment (AC1) · 22 Feb 2018

Please, find our response to the referee's comment and an updated version of our manuscript in the zip archive in supplement to this message.

Please also note the supplement to this comment: https://www.biogeosciences-discuss.net/bg-2017-391/bg-2017-391-AC1-supplement.zip

---

## Author Comment (AC2) · 22 Feb 2018

Please, find our response to the referee's comment as well as an updated version of our manuscript in the supplement to this message.

Please also note the supplement to this comment: https://www.biogeosciences-discuss.net/bg-2017-391/bg-2017-391-AC2-supplement.zip
* * *

---

## Author Response (AR1)

**Anonymous Referee #1 (R1)**

R1: The authors have conducted a very nice study that works through uncertainties inherent in the global uptake of CO2 associated with wind forcing uncertainties. The presentation is clear, the methods are transparent, and the results should be of broad interest to the ocean carbon cycle community. The only shortcoming of the study is that it is somewhat technical, and in order to satisfy the scientific priorities of Biogeosciences my recommendation would be that the manuscript would benefit from minor revisions before being accepted for publication. A number of more general and more specific questions/comments are raised in the text that follows.

Author's response: We are grateful for the reviewer's evaluation and his/her constructive suggestions. Please find below a detailed answer to each comment.

On behalf of all co-authors,
Alizée Roobaert

R1: First and foremost, in the discuss pertaining to Fig. 2 and Fig. 3 and Fig. 4, it would be useful if the authors could identify whether they authors see pertinent dynamical or circulation structures as dominating the uncertainties. Given the linear color scale in Fig. 3 and Fig. 4, it appears that the dominant uncertainties might be mostly over western boundary current regions? This would appear to be important, as there is currently a question being discussed in the carbon cycle community of whether western boundary currents serve as hot spots for carbon uptake. It would be of wide interest if the authors could attempt to quantify a response to the following question: Even with a perfect observing system for perfect pCO2 measurements that is seasonally-resolving, what inherent uncertainties from gas exchange parameterizations and winds apply over these particular structures? This is already implicitly there in the text, but it would be helpful to emphasize this a bit to make clear what challenges lie ahead for more regionals-focused mechanistic interpretations.

Author's response: We agree with the reviewer that associating our uncertainties in $FCO_2$ with large dynamical or circulation structures would be relevant to our manuscript. However, our analysis of the spatial distribution of $FCO_2$ using different wind products did not allow clearly identifying such connection beyond the broad stroke picture already provided in our discussion. This motivated us to perform a regional scale analysis using RECCAP regions in our discussion. It is true however that several Western Boundary Current regions (in particular Brazil Current/Malvinas Current and the Florida Current) tend to display relatively large uncertainties in our calculations. Considering the particular interest for these regions in terms of $CO_2$ exchange with the atmosphere (Cai 2011; Gruber et al. 2009; Laruelle et al. 2010, 2014), we discussed briefly these regions in our revised manuscript (see updated text in bold below). We also introduced a cautionary statement regarding upwelling regions because the spatial extent of our $pCO_2$ product does not resolve well nearshore coastal regions and might thus miss part of the Eastern and Western Boundary Currents.

*"The discrepancies between $FCO_2$ generated using NCEP2 and those generated using the other wind products are particularly pronounced near the equator, in the Arctic region and around 40° S (Austral Ocean) and 40° N (Fig. 2a and Fig. 3). For example, at these mid-*

*latitudes in the north and south hemisphere, differences between $FCO_{2-NCEP1}$ and $FCO_{2-NCEP2}$ can reach 0.8 and 0.6 mol C $m^{-2}$ $yr^{-1}$, respectively. Such pronounced differences result from the combination of relatively high wind speeds and significant $pCO_2$ gradients (> 25 µatm) as well as significant discrepancies between NCEP1 and NCEP2 at these latitudes (Fig. 2b).* **_Other regions characterized by large differences in $FCO_2$ depending on the applied wind product include western boundary currents such as the Brazilian/ Malvinas Current and the Florida Current, which generally are regions of intense $CO_2$ outgassing (Cai, 2011; Laruelle et al., 2010, 2014). It should be noted, however, that the spatial extent of our $pCO_2$ data product does not include the near coastal zone and thus only partly cover these areas._** *Comparing the air-sea $CO_2$ exchange using all climatological mean wind products, we find that CCMP (global wind average of 7.5 m s-1 from 1991 through 2011, which is close to that calculated by Wanninkhof (2014) for the period 1990-2009 of 7.3 m $s^{-1}$) leads to a slightly more intense $CO_2$ exchange between 40° S-40° N and in the Arctic region (> 60° N) than $FCO_{2-ERA}$ and $FCO_{2-NCEP1}$ (Fig.2a)."*

R1 : It would also be beneficial for anchoring the present study in the published literature if the authors could relate their results to the study of Sarmiento, Orr, and Siegenthaler (1992; JGR), where it was reported within a modeling context that 100 % increases in gas exchange only impact 9.2 % changes in air-sea CO2 fluxes. Although that study from 25 years ago used a simplified representation of anthropogenic carbon, it has long been cited for the argument that gas exchange representation isn't of critical importance for global uptake.

Author's response:

We agree with the reviewer that it is important to discuss the implications of the uncertainties in $FCO_2$ calculated in our study for the global oceanic $CO_2$ in a modelling context, and to refer to the Sarmiento's paper. We think, however, that it is difficult to translate, in a quantitative way, our uncertainties in the $FCO_2$ calculation framework to global scale models for several reasons:

With respect to the uncertainties associated with the k-parametrization alone in a modelling context, Sarmiento, Orr, and Siegenthaler (1992) used a three-dimensional global general circulation model to demonstrate that the sensitivity of the globally integrated anthropogenic uptake of $CO_2$ by the ocean to the formulation of k is relatively low. This is partly due to the feedback mechanisms in these models: for instance, a doubling of the mean k value will induce an increase of the anthropogenic air-sea exchange in some regions (i.e. in the polar and equatorial regions). Because of the dynamic nature of a global circulation model, which recalculates the values of its stocks and fluxes at each time step, $FCO_2$ will increase but not proportionally to the doubling of k since the air-sea $pCO_2$ gradient will decrease when k is larger, acting as a negative feedback on $FCO_2$ changes. As shown by Sarmiento et al (1992), the overall effect of doubling k leads to only about a 10 % change in anthropogenic $CO_2$ absorption by the ocean.

In our study, the air-sea $CO_2$ exchange calculations directly derive from observations and do not have this air-sea gradient adjustment feedback mechanism, which compensates for the doubling of k. A direct comparison of uncertainties in quantitative terms is thus not straightforward. Nonetheless, our observation-based analysis indicates that uncertainties are comparatively higher in some regions (mainly polar and equatorial regions) and as shown in Fig 13 of Sarmiento's paper, broadly similar patterns can be diagnosed in a modelling context. Furthermore, the formulation of k in Sarmiento's paper is linear with respect to wind

speed, thereby underestimating the effect of strong winds on the $CO_2$ gas exchange. Thus, we expect a higher sensitivity of global $FCO_2$ to changes in k when a quadratic formulation is used, as is the case in our study. Finally, in the Sarmiento's paper, the uncertainties do not take into account the influence of the choice of one wind product over another. As shown by the study of Ishii (2014) for the Pacific Ocean, significant $FCO_2$ differences can be observed using the same model but different wind products.

We added several sentences to our manuscript to reflect on the implication of our findings for the parametrization of the $CO_2$ exchange with the atmosphere in global oceanic models and refer to the study of Sarmiento et al. (1992) for context (see updated text in bold below).

*"Our calculations reveal that, whenever a formulation of k is used to quantify the global oceanic $FCO_2$ indistinctly with ERA, CCMP or NCEP1, the range of estimates will be associated with an uncertainty of the order of 12 % when combined with recent global formulation of k derived from the $^{14}C$ global inventory, only. This uncertainty significantly rises when using the out dated formulation proposed by Wanninkhof (1992), a hybrid k-formulation (Wanninkhof et al., 2009) and/or when $FCO_2$ is calculated with NCEP2. Furthermore, our results have highlighted that due to differences in the regional wind patterns, regional discrepancies in $FCO_2$ are even larger than global. Finally, other poorly constrained sources of uncertainty in the calculation of $FCO_2$ and not included in our study exist in polar and coastal regions when specific processes further complicate the air-water exchange. For instance, in partially ice covered areas, the relationship between the intensity of the gas exchange is more complex than a direct linear scaling to the ice-free surface area (Lovely et al., 2015), but no generic formulation exists yet to account for this effect. Similarly, in some coastal areas, specific physical processes such as the occurrence of surfactants or other sources of turbulences than wind such as tidal currents may affect the intensity of the exchange of $CO_2$ at the air-water interface (Ho et al., 2011). In the future, the quantification of the effect of such processes on the uncertainty over the air-water $CO_2$ exchange will have to be further investigated to better constrain regional carbon budgets.* __It should be noted that it is difficult to directly extrapolate our results to $FCO_2$ derived from global circulation models and Earth System Models. Indeed, because of the dynamic air-sea $pCO_2$ gradient adjustment acting against the change in gas transfer velocity in these models, the effect of variations in k on global $FCO_2$ estimates are dampened. For instance, Sarmiento et al. (1992) showed that a doubling in k resulted in only about a 10 % increase in the overall anthropogenic $CO_2$ absorption by the ocean. Because of the absence of this negative feedback mechanism in observation-based estimates, it is expected that wind-induced uncertainties derived from observations will be larger than uncertainties derived from Ocean general circulation models (OGCMs) and Earth System Models. Furthermore, the use of a linear k formulation and a single wind product in Sarmiento et al. (1992) will lead to smaller uncertainties than in our assessment based on quadratic formulations and multiple wind products. As shown by the results of Ishii (2014) for the Pacific Ocean, significant $FCO_2$ differences can be observed using the same model but different wind products.__ *"*

R1: In a related point, it would also be beneficial if the authors could likewise relate the uncertainties here to those reported in the Rödenbeck et al. (2015) comparison of fluxes found for different gridded pCO2 products from different global research groups.

Author's response: Our research focused on the effect of the formulation of k and the choice of the wind product on the uncertainty in $FCO_2$. We thus did not include the choice of the $pCO_2$ product in our analysis but we agree that comparing the range of global $FCO_2$ obtained with a given $pCO_2$ product and different wind products with the range of global $FCO_2$ obtained with a single formulation for the $CO_2$ exchange but several $pCO_2$ mapping

techniques, would be relevant to our manuscript. In Rödenbeck's study estimates of the global $FCO_2$ calculated with the same parametrization of k but different $pCO_2$ products range from -1.36 Pg C $yr^{-1}$ to -1.96 Pg C $yr^{-1}$. Such a difference (~30 %), using 14 different $pCO_2$ data products is larger than the range obtained in our study using different formulations of k and all wind products but NCEP2 (20 %). Following the reviewer's advice, we added several sentences to refer to Rödenbeck's study in our manuscript and compare our uncertainties related to the formulation of k to those associated with the $pCO_2$ mapping technique (see updated text in bold below).

"This study indicates that a change in spatial resolution of the wind data from 4° x 5° degrees to 1° x 1° using monthly winds leads to discrepancies in c values of about 3 % while the change in temporal resolution from daily to monthly using a 1° x 1° spatial resolution also leads to an uncertainty of about 3 %. ***Furthermore***, as already pointed by Wanninkhof (1992), the use of monthly averaged values of $U_{10}$ instead of the 6 hour $$ has a much bigger effect on $FCO_2$, with an underestimation reaching over 20 %. ***It is also interesting to compare our reported $FCO_2$ uncertainties to those introduced by the choice of a given $pCO_2$ product. The application of distinct interpolation techniques in recent years has led to the publication of several global $pCO_2$ products that are largely based on the same observational dataset (i.e. SOCAT, Bakker et al., 2016). To quantify the uncertainty introduced by the choice of the pCO2 field, Rödenbeck et al. (2015) applied an identical parameterization of the $CO_2$ exchange at the air-water interface to 14 $pCO_2$ data products. The global $FCO_2$ ranged from -1.36 Pg C $yr^{-1}$ to -1.96 Pg C $yr^{-1}$, and the relative difference (~30 %) is thus slightly larger than the one attributed to different formulations of k and wind products (20 %, ignoring NCEP2) calculated here.***"

R1: The authors point out that there are important discrepancies between global and empirical formulations of gas exchange. It could be constructive in this regard to point out that there are expected to be important limitations with the construct of a piston velocity in representing the relationship between surface carbon fluxes and winds. Winds don't only impact air-sea fluxes through microturbulence at the air-sea interface, but in the sense of climate dynamics the very same winds can in some regions sustain entrainment of waters through deepening of the mixed lawyer (through shear-induced turbulence). As the relative amplitude of these processes is not known, one should expect that parameterizations based on the concept of piston velocity (the first of these) will have limitations (Rodgers et al., 2014, BG), and that parameterizations based on local properties at the air-sea interface may thereby have inherent uncertainties that are irreducible. I think that this point in fact strengthens the main arguments in the manuscript, in that systematic and rigorous analysis of uncertainties will continue to be a critical component of carbon cycle research moving forward. Regarding the first of the points above (western boundary currents), this could be pertinent.

Author's response: We agree with the reviewer that, for local parameterization, the influence of wind on the gas exchange is not limited to its influence on the piston velocity. As the reviewer points out, Rodgers et al. (2014) quantified the influence of wind stirring on the $CO_2$ exchange at the air-water interface in the Southern Ocean. In that study, the reduction of the exchange of $CO_2$ at the air-water interface is partly controlled by the effect of wind stirring on the depth of the mixed layer. This contributes to the complexity of deriving an accurate formulation of k. We modified the last section of our discussion (see updated text in bold below) to include a reference to Rodgers et al. (2014) and account for those points mentioned by the reviewer.

*"This further supports the idea that empirical formulations are calibrated for specific local settings and are not suitable for global scale applications. For instance, the Kuss et al. (2004) and the Weiss et al. (2007) relationships were derived in areas of the Baltic Sea characterized by very high wind speeds, up to 20 m s$^{-1}$. **In addition, locally, wind may influence the intensity of the $CO_2$ exchange at the air-water interface by other processes not connected to the turbulence at the interface and the piston velocity. Rodgers et al. (2014), for example, identified the effect of wind speed as a control of $FCO_2$ in the Southern Ocean through its control on the depth of the mixed layer depth through wind stirring. This kind of indirect controls of wind on the $CO_2$ exchange at the air-water interface adds an additional important source of uncertainty on local parameterization of k.**"*

R1: Minor Comments:
pg. 2, line 5 "open" should be "opened"
Author's response: Done
pg. 2, line 12: "observation-based" should probably be "observationally-based"
Author's response: Done
pg 2 line 19 should be "10 meters"
Author's response: Done
pg 2 line 28: should be "uncertainty associated with"
Author's response: Done
pg 3 lines 3-4 should be "the latitudinal distribution of FCO2"
Author's response: Done
pg 3 line 10 I recommend removing "in a nutshell" (rather informal)
Author's response: Done
pg 3 line 14 change "measure because z" to "measure as z"
Author's response: Done
pg 4 line 12 again, I think this should say "observationally-based"
Author's response: Done
pg 4 line 18 I think this should say "using a two-step"
Author's response: Done
pg 4 line 19 should say "maps for the global ocean"
Author's response: Done
pg 7 line 17 should say "Within the tropics"
Author's response: In this sentence, we refer to the latitudes close to 23 degrees South and North but not the entire area in between. We think that using 'Within the tropics' instead of along the tropics might be misleading to the reader.
pg 9 line 21 should say "This corresponds"
Author's response: Done
pg 9 line 22 should say "despite the fact that these empirical"
Author's response: Done
pg 11 line 16 replace "the literature also reports" with "one also finds in the published literature"
Author's response: Done
pg 11 line 24 should say "methods have their relative advantages"
Author's response: Done
pg 11 line 24 should say "has been shown that"
Author's response: Done

Summary Statement:
To restate, I believe that with some relatively minor text changes that connect the present study to broader community efforts and scientific interests, the manuscript should meet the standard for publication in Biogeosciences. The authors are to be

commended for a very nice and thorough analysis and presentation that will be of brand interest.

We thank again the reviewer for his/her constructive remarks and his/her support of our study.
R2 : In their manuscript the authors present an in depth study of the impact of different gridded wind speed data products - such as CCMP, ERA, NCEP1 and NCEP2 – utilized to determine air sea fluxes of carbon dioxide on global and regional scales. Therefore they combined those data with a globally re-gridded sea surface climatology of pCO2 values covering the years from 1991 to 2011. By employing different parameterizations for the calculation of CO2 fluxes they found a strong dependence of this number on the choice of the specific wind speed product.
To constrain the variability of air to sea fluxes of CO2 is of great importance for assessing the global ocean carbon sink and the concomitant acidification. In order to minimize the error in the flux calculations the authors propose to recalibrate the piston velocities (k-formulations) for the respective wind speed data product.
The manuscript is well written and contains important informations and innovations.
The methods and results are clearly presented. However, the manuscript would benefit from a short discussion of the consequences for Earth system modeling. What is the expected impact of the findings in this study on model projections regarding the evolution of the future carbon sink? A short clarifying paragraph would be helpful.

I recommend publication in "Biogeosciences"

As elaborated in answer two of reviewer 1, it is not straightforward to compare the observation-based uncertainties derived here to model-derived uncertainty assessments. Nevertheless, as explained in the revised text (pg 15 lines 2-10), one can speculate that because of the dynamic $pCO_2$ adjustment mechanism in models, combined with the use of a single k formulation and a given wind product to carry model projections, the uncertainties induced by the wind forcing could be underestimated in the assessment of the future ocean carbon sink by a given model.

Uncertainties in the present and future ocean $CO_2$ sink are constrained by model ensembles relying on different k parameterizations, wind products and spatio-temporal resolutions. But as already mentioned in the original manuscript "*In numerous modeling studies, FCO$_2$ is still calculated using k-parametrizations from the literature combined with a different wind product from the one used to calibrate the coefficient c (e.g. Aumont and Bopp, 2006; Bourgeois et al., 2016; Matear and Lenton, 2008; Le Quéré et al., 2007; Schwinger et al., 2016; Thomas et al., 2008)* ». Following reviewer's 2 suggestion, we have now added in the revised text (pg 10 lines 26-28): « ***These inconsistencies call into question the assessment of the wind-induced uncertainties associated with the future ocean CO$_2$ sink and a systematic approach, similar to the one used for the observation-based estimation of the present-day FCO$_2$, should help better constrain model-derived uncertainties.*** »

As well as some considerations regarding the need for an update of these formulations of k in global models at the end of our conclusions section (pg 15 lines 11-17):
"***Currently, the uncertainty in the global CO$_2$ uptake by the ocean is estimated by comparing multiple global models (Ciais et al., 2013). Unfortunately, these models use various formulations of k, some of which are outdated like that of Wanninkhof (1992)***

*and wind products which are not always consistent with their formulation of the piston velocity. Based on our analysis of the impact of the choice of the wind product and its resolution on $FCO_2$, we believe it would be beneficial to update these representations of the $CO_2$ exchange in these models. Ideally, the value of c should be adapted to match a global k consistent with the global average derived from the latest $^{14}C$ budget (Wanninkohf et al., 2014).*"

Finally, models are constantly evaluated and improved upon by observations based estimates, for the calculation of which our uncertainty estimates are relevant. We believe that it is in this context that our study is also particularly relevant and potentially the most impactful for the global carbon modelling community and future model projections.

Again, we are grateful for the reviewer's evaluation of our manuscript.
On behalf of all co-authors,
Alizée Roobaert